# SPLITTED WAVELET DIFFERENTIAL INCLUSION

## ABSTRACT

Wavelet Shrinkage typically selects only a small proportion of large coefficients via soft or hard thresholding, since the *strong signal* composed by these coefficients has more semantic meaning than others. Typical examples include the object's shape in the image or the burst activity in the low $\beta$ band in Parkinson's Disease. However, it has been found that there also exists *weak signal* that should not be ignored. Such a weak signal refers to the set of small coefficients, which in the above examples *resp.* correspond to the texture of an image or the non-burst/tonic activity in Parkinson's Disease. Although it is not as interpretable as the strong signal, ignorance of it may miss information in signal reconstruction. Existing methods either suffered from failing to disentangle the strong signal apart with a too small threshold parameter, or inaccurate estimation of the whole signal (*i.e.*, strong and weak signals) due to the bias/errors in the strong signal and over-smoothing of the weak signal. To resolve these problems, we propose a *Splitted Wavelet Differential Inclusion*, which is provable to achieve better estimation on both the strong signal and the whole signal than Wavelet Shrinkage. Specifically, equipped with an $\ell_2$ splitting mechanism, we obtain the solution path from the differential inclusion of a couple of parameters, of which the sparse one can remove bias in estimating the strong signal and the dense parameter can additionally capture the weak signal with the $\ell_2$ shrinkage. The utility of our method is demonstrated by the improved accuracy in a numerical experiment and moreover the additional findings of tonic activity in Parkinson's Disease.

## 1 INTRODUCTION

Wavelet Shrinkage Donoho & Johnstone (1994; 1995; 1998); Donoho (1995), which has been widely applied in image denoising Goyal et al. (2020), electrocardiogram (ECG) data denoising Priya et al. (2016) and brain fMRI (Functional magnetic resonance imaging) data analysis Wang et al. (2015), projected the noisy data into the Wavelet domain, followed by a hard or soft thresholding method to force noisy coefficients to zeros. To select the threshold parameter, many methods have been proposed, such as *Universal* ($\sigma\sqrt{2\log n}$) Donoho & Johnstone (1994), Minimaxi Verma & Verma (2012), *SureShrink* Donoho & Johnstone (1995), Bayesian Shrinkage Do & Vetterli (2002); Simoncelli & Adelson (1996); JOHNSTONE & SILVERMAN (2005), non-parametric shrinkage Antoniadis & Fan (2001); Gao (1998) such as SureLet Luisier et al. (2007) and Neigh Shrink Sure Chen et al. (2005).

Most of these methods only selected a small proportion of large coefficients since they are assumed to be all the information contained in the signal Atto et al. (2011). However, it has been found in many applications that there may exist small coefficients Donoho & Johnstone (1994), such as texture/contour in image denoising Atto et al. (2011), the non-burst activity in Parkinson's DiseaseKhawaldeh et al. (2020), procedural bias/enlarged gray matter voxels in Alzheimer's Disease Sun et al. (2017). In this paper, the signal composed by such small (*resp.* large) coefficients is called the weak (*resp.* strong) signal. Although the strong signal typically has more semantic meaning and is more interpretable than the weak signal, ignorance of the latter can lead to the loss of information in reconstruction. For example, although elevated energy in the $\beta$ band (8-35 Hz) of subthalamic nucleus local field potentials is a significant biomarker in the Parkinson's study, the non-burst activity was also found to contain more information on motor prediction Khawaldeh et al. (2020); Brittain & Brown (2014); Mallet et al. (2008). Therefore, it is desired to identify the strong signal as a semantic part and meanwhile estimate the whole signal (including both strong and weak signals) well.

For these tasks, existing methods with too small or too large thresholds either suffer from failing to eliminate noise components in estimating the strong signal or ignoring the weak signal part in estimating the whole signal. Specifically, the methods with too large threshold values (*e.g.*, Universal, or Minimaxi) can eliminate the noise components, but they can induce large bias/errors in estimating the strong signal; moreover, they can over-smooth the weak signal part with small coefficients. On the other, the methods with smaller thresholds (*e.g.* SureShrink Donoho & Johnstone (1995)) may fail to eliminate noise components in estimating the strong signal for interpretation.

To resolve these problems, we in this paper propose a differential inclusion method with an $\ell_2$ splitting mechanism, dubbed as *Splitted Wavelet Differential Inclusion*, which is provable to achieve better estimation than Wavelet Shrinkage on both the strong signal and the whole signal. Specifically, we can obtain a regularized solution path from the differential inclusion of a couple of sparse and dense parameters introduced via the $\ell_2$ splitting mechanism, where the sparse parameter can estimate the strong signal without bias via early stopping while the dense parameter can additionally capture the weak signal via $\ell_2$ shrinkage. We can show that our method has a unique closed-form solution path. To extend to a non-orthogonal Wavelet Decomposition, we provide a discretization method that can effectively generate the solution path. The utility and effectiveness of our method are demonstrated in a numerical experiment and the effectiveness study of Dopaminergic medication in Parkinson's Disease. Particularly, the signal recovered by our method is more significantly correlated to the medication, which can be explained by the non-burst activity found by our method.

To summarize, our contributions are listed as follows:

- We propose the *Splitted Wavelet Differential Inclusion*, which involves a dual parameter to simultaneously estimate the strong and the whole signal.

- We theoretically show that the closed-form solution path can achieve better estimation than Wavelet Shrinkage on both the strong signal and the whole signal. For the strong signal, our estimation is bias-free while for the whole signal, our dense parameter can accurately capture the weak signal via an $\ell_2$ shrinkage.

- We apply our method to neural signal recognition in Parkinson's Disease and identify the non-burst activity that has been recently found to be also responsive to the medication.

## 2 RELATED WORK

**Wavelet Shrinkage.** The threshold selection has been a challenging problem for Wavelet shrinkage methods, which can be traced back to Donoho & Johnstone (1994) that proposed a universal selection parameter $\sigma\sqrt{2\log n}$ where $n$ denotes the length of the signal and $\sigma$ denotes the noise level. Although it can eliminate noise components, it can induce biases/errors in estimation. To resolve this problem, many non-adaptive and adaptive methods have been proposed, such as Minimaxi Verma & Verma (2012), *SureShrinkage* Donoho & Johnstone (1995) that leveraged the Jame-Stein Shrinkage method for more accurate estimation, Bayesian shrinkage Do & Vetterli (2002); Simoncelli & Adelson (1996); JOHNSTONE & SILVERMAN (2005); ter Braak (2006), non-parametric shrinkage Antoniadis & Fan (2001); Gao (1998) including SureLet Luisier et al. (2007) and Neigh Shrink Sure Chen et al. (2005), *etc*.

Some methods above relied on the *strongly sparsity* assumption Atto et al. (2011), *i.e.*, the signal (or each sub-band) is a representation of only a small proportion of strong signal coefficients with large magnitude. As introduced next, there may exist small/weak coefficients in many applications (such as image denoising, and neuroimaging analysis) and should not be ignored.

**Weak Signal Coefficients** refer to those small coefficients that have been found in many applications, such as textures, contours in image denoising Atto et al. (2011), procedural bias/enlarged gray matter voxels in Alzheimer's Disease Sun et al. (2017), the non-burst component in dopamine-dependent motor symptoms with Parkinson's patients Khawaldeh et al. (2020); Brittain & Brown (2014); Mallet et al. (2008). In these applications, these weak signal coefficients may not be as interpretable/of semantic meaning as strong signal coefficients (*e.g.*, lesion voxels that are pathologically related to Alzheimer's Disease or burst component in Parkinson's Disease analysis); however, the ignorance of these weak signals due to over-smoothing may lose information in image reconstruction or prediction power in disease diagnosis. Therefore, it is desired to i) disentangle the strong

signals as the semantic component of the signal, and meanwhile ii) accurately estimate the whole signal by capturing the weak signal coefficients.

**Limitations of Wavelet Shrinkage and Our Specifications.** Existing Wavelet Shrinkage methods either suffer from failing to disentangle strong signals apart because of small thresholds (*e.g.*, SureShrink), or ignoring weak signals because of large thresholds (*e.g.*, Minimaxi or Universal $\sqrt{2 \log n}$). **In contrast**, the splitting mechanism in our *Wavelet Differential Inclusion* introduces a couple of sparse and dense parameters, which can respectively estimate the strong signal without bias and the whole signal by capturing weak signal coefficients. Note that our method is motivated by but different from the differential inclusion method in signal recovery Osher et al. (2005; 2016). For the reason of coherence and space limit, we leave the review of these methods in Appx. A.

## 3 PRELIMINARY

**Problem Setup.** Suppose we observe data $\{y_i\}_{i=1}^n$ with $n = 2^J$, such that $y_i = f(t_i) + e_i$ with $e_1, ..., e_n \sim_{i.i.d} \mathcal{N}(0, \sigma^2)$, $t_i := \frac{i}{n}$ and $f$ denoting the ground-truth signal we would like to recover. One may construct a wavelet transformation to decompose the observed signal $y$ into an orthogonal wavelet basis, including *stationary wavelet transform* (SWT), *discrete wavelet transform* (DWT), *etc.* For discrete wavelet transform, we can obtain a wavelet matrix $W \in \mathbb{R}^{n \times n}$ depending on the type of wavelet filters (such as Coiflets, Symlets, Daubechies Cohen et al. (1993), Beylkin Beylkin et al. (1991), Morris minimum-bandwidth Morris & Peravali (1999)), the number of vanishing moments $M$ and the coarsest resolution level $L$.

With such $W$, we can obtain the coefficients $\theta$ up to the linear transformation of noise:

$$\omega = \theta^* + \varepsilon, \ \omega = Wy, \ \varepsilon = We \sim \mathcal{N}(0, \sigma^2 I_n),$$

when $W$ is orthogonal. Here, we assume the $\theta^*$ contains three types of coefficients:

1. *Strong Signal Coefficients.* $S := \{i : |\theta_i^*| > 2\sigma(1 + a)\sqrt{2 \log n}\}$ for some constant $a > 0$.
2. *Weak Signal Coefficients.* $T := \{i : 1 < |\theta_i^*| = o(\sqrt{2 \log n})\}$.
3. *Null.* $N := \{i : \theta_i^* = 0\}$.

**Wavelet Shrinkage via Soft-Thresholding.** The Wavelet Shrinkage method in Donoho & Johnstone (1994); Donoho (1993); Donoho & Johnstone (1995) proposed the soft-thresholding estimator $\hat{\theta}_i = \eta(\omega_i, \lambda) = \text{sign}(\omega_i) \max(|\omega_i| - \lambda, 0)$ for some $\lambda > 0$, followed by inverse wavelet transformation to recover the signal $\hat{f} := W^{-1}\eta(\omega, \lambda)$. To remove the noise components, the Donoho & Johnstone (1994) selected $\lambda \sim O(\sqrt{2 \log n})$, which is provable to be minimax optimal.

**Proposition 3.1** (Theorem 2 in Donoho & Johnstone (1994)). *Denote $\hat{\theta}(\lambda_n) := \eta(\omega, \lambda_n)$, then the minimax threshold $\lambda_n^*$ is $\lambda_n^* := \arg\inf_{\lambda \geq 0} \sup_{\theta^*} \mathbb{E}[\|\hat{\theta}(\lambda_n) - \theta^*\|_2^2] \sim \sqrt{2 \log n}$.*

Although $\lambda \sim \sqrt{2 \log n}$ can effectively remove noise with high probability, it suffers from two limitations: **i)** the estimation of strong signal coefficients is biased due to non-zero $\lambda$; **ii)** it ignores the weak signal coefficients, which leads to additional errors in estimating the whole signal $\theta^*$. Specifically, for **i)**, although the threshold $\sqrt{2 \log n}$ can identify strong coefficients $S$ by removing others with high probability, it can induce bias in estimating $\theta^{*,s}$. This is shown by the following result, which states that once we identify $S$, the optimal threshold value in estimating $\theta^{*,s}$ is $\lambda = 0$.

**Proposition 3.2.** *We have $0 = \arg\inf_\lambda \sup_{|\theta_i^*| \geq 1} \mathbb{E}(\eta(\omega_i, \lambda) - \theta_i^*)^2$.*

*Remark* 3.3. This result shows that in terms of population error, the best optimal threshold value is also 0 for weak coefficients. However, it does not mean we should select $\lambda = 0$ to estimate $\theta^*$. First, it fails to remove noise components in $N$. Moreover, even for weak signals, we will show that with an appropriate non-zero, we can achieve better estimation with a high probability.

Besides for **ii)**, the $\lambda \sim \sqrt{2 \log n}$ fails to account for weak signal coefficients that are $o(\sqrt{2 \log n})$. To achieve a more accurate estimation, the Donoho & Johnstone (1995) proposed *SureShrink* (Stein's unbiased estimate of risk), which can reduce biases in $\theta^*$. However, it can mistakenly induce noise and weak signals into estimation, thus failing to disentangle the strong signal (interpretable part) apart. This may be undesired in many applications, *e.g.*, bust component identification in Biomarker identification in Parkinson's patients, or lesion features identification in Alzheimer's Disease.

In summary, existing methods either suffer from bias of strong signal coefficients and ignorance of weak coefficients in estimating $\theta^*$; or fail to disentangle the strong signal $\theta^{*,s}$ apart.

## 4 SPLITTED WAVELET DIFFERENTIAL INCLUSION

We introduce a new method from the perspective of *differential inclusion*, which can simultaneously identify the strong signal $\theta^{*,s}$ (in Sec. 4.1) and accurately estimate the whole signal $\theta^*$ (in Sec. 4.2).

### 4.1 WAVELET DIFFERENTIAL INCLUSION FOR THE STRONG SIGNAL $\theta^{*,s}$

In this section, we introduce our method to remove the bias in estimating the strong signal $\theta^{*,s}$, by considering the following differential inclusion:

$$\dot{\rho}(t) = -\nabla_\theta \ell(\theta(t)) = \omega - \theta(t), \tag{1a}$$

$$\rho(t) \in \partial \|\theta(t)\|_1, \tag{1b}$$

where we denote $\ell(\theta) := \frac{1}{2}\|\omega - \theta\|_2^2$ and $\rho(0) = \theta(0) = 0$. We call Eq. 1 as *Wavelet Differential Inclusion* (WDI). Note that it is a special form of the *Bregman Inverse Scale Space* (ISS) Osher et al. (2016) when the design matrix is set to $I_n$ in the linear model, which can be viewed as a continuous dynamics of *Bregman Iteration* proposed to Osher et al. (2005) in image denoising. Thanks to the identity matrix design, the differential inclusion of Eq. 1 in the Wavelet scenario has a closed-form solution $\theta(t)$, which can illustrate the effectiveness of bias removal in estimating $\theta^*$.

Specifically, starting from $\rho(0) = 0$, Eq. 1 generates a unique solution path of $\theta(t)$, in which more elements become non-zeros as $t$ grows as shown in the following proposition:

**Proposition 4.1.** *The solution of differential inclusion in Eq. 1 is $\theta_j(t) = \omega_j(t)$ for $t \geq \frac{1}{|\omega_j|}$; and $= 0$ otherwise for each $j$.*

*Remark* 4.2. To explain the effect of bias removal, we compare Eq. 1 with Wavelet Shrinkage, whose solution $\theta(t)$ $(t = \frac{1}{\lambda})$ is equivalent to the Lasso estimator: $\frac{1}{2}\|\omega - \theta\|_2^2 + \frac{1}{t}\|\theta\|_1$, whose solution satisfies $\frac{\rho(t)}{t} = \omega - \theta(t)$ with $\rho(0) = \theta(0) = 0$. When $|\rho_j(t)|$ becomes non-zero, then we have $\theta_j(t) = \omega_j - \frac{\rho_j(t)}{t}$, where $\frac{\rho_j(t)}{t}$ can induce the bias. As a contrast, when $|\rho_j(t)| = 1$, its gradient $\dot{\rho}_j(t) = 0$, which can give $\theta_j(t) = \omega_j$ and hence can remove the bias.

As shown in Prop. D.1, $t$ plays a similar role as $1/\lambda$ in disentangling the strong signal from others. However, it is interesting to note that different from the Wavelet Shrinkage, the solution $\theta_j(t) = \omega_j = \eta(\omega_j, \lambda = 0)$ is without additional threshold parameter! In contrast to $\theta_j(\lambda = \sqrt{2\log n}) := \eta(\omega_j, \sqrt{2\log n})$, this estimator can remove the bias caused by $\lambda$. Therefore, our differential inclusion can not only remove noise and weak components when $t$ is large enough but also can estimate $\theta^{*,s}$ without bias induced by $\lambda$ that is necessary for removing bias in Wavelet Shrinkage. Equipped with bias removal of Eq. 1, we can achieve a smaller $\ell_2$ error than the Wavelet Shrinkage.

**Theorem 4.3.** *Denote $\theta_{\min}^{*,s} := \min_{i \in S} |\theta_i^*|$ and $s := |S|$. Then at $\bar{\tau} := 1/((1+a)\sqrt{2\log n})$ and $n$ is large enough such that $\frac{a}{2}\sqrt{2\log n} > \theta_j^*$ for $j \notin S$. Then with probability at least $1 - 2n^{-4a^2} - \max\left(\exp\left(-s\lambda^2/8\right), n^{-(1+a)^2/4}\right)$, we have*

$$\|\theta(\bar{\tau}) - \theta^{*,s}\|_2 < \|\eta(\omega, \lambda) - \theta^{*,s}\|_2 \ \forall \lambda > 0, \text{where } \theta(t) \text{ is the solution of Eq. 1.} \tag{2}$$

*Remark* 4.4. The proof of Thm. 4.3 is left to Appx. D. We will show that the $\bar{\tau}$ can disentangle $S$ from other components since $\max_j |\varepsilon_j| < (1+a)\sqrt{2\log n}$ with high probability.

### 4.2 WAVELET DIFFERENTIAL INCLUSION WITH $\ell_2$-SPLITTING FOR THE WHOLE SIGNAL $\theta^*$

In this section, we proceed to introduce our method to additionally capture the weak signal, so as to well estimate the whole signal $\theta^*$. To achieve this goal, we propose the *Splitted Wavelet Differential Inclusion* (SWDI), which generates a solution path of $\theta^s(t) \in \mathbb{R}^n$ (to estimate $\theta^{*,s}$) coupled with a dense parameter $\theta(t)$ (to estimate $\theta^*$) introduced by an $\ell_2$ splitting term. We will show that the $\theta^s(t)$ maintains the same bias removal property as WDI in Eq. 1; moreover, the dense parameter can additionally capture the weak signal with the $\ell_2$ shrinkage induced by the $\ell_2$ splitting mechanism.

Specifically, we consider the loss $\bar{\ell}_\rho(\theta, \theta^s) := \frac{1}{2}\|\omega - \theta\|_2^2 + \frac{\rho}{2}\|\theta - \theta^s\|_2^2$, where $\frac{\rho}{2}\|\theta - \theta^s\|_2^2$ with $\rho > 0$ denotes the $\ell_2$ splitting term that introduces a couple of parameters, which is expected to simultaneously estimate the strong signal $\theta^{*,s}$ and the whole signal $\theta^*$ well. This is achieved by the following differential inclusion:

$$0 = -\nabla_\theta \bar{\ell}_\rho(\theta, \theta^s) = \omega - \theta(t) + \rho(\theta^s(t) - \theta(t)), \tag{3a}$$

$$\dot{\rho}(t) = -\nabla_{\theta^s} \bar{\ell}_\rho(\theta, \theta^s) = \rho(\theta(t) - \theta^s(t)), \tag{3b}$$

$$\rho(t) \in \partial\|\theta^s(t)\|_1, \tag{3c}$$

where $\rho(0) = \theta(0) = \theta^s(0)$. Similar to the WDI in Eq. 1, when $t$ is large enough, it can remove the noise and weak signal components, *i.e.*, $T \cup N$ to identify the strong signal component in $\theta^s$; while the $\theta$ is always dense, *i.e.*, all elements are non-zeros, which can hence additionally capture the weak components. Formally, Eq. 3 can generate an unique solution path:

**Proposition 4.5.** *The solution of differential inclusion in Eq. 3 is*

$$\begin{cases} \theta_j(t) = \theta_j^s(t) = \omega_j, & t \geq (1 + 1/\rho)/(\omega_j) \\ \theta_j(t) = \frac{\omega_j}{1+\rho}, \theta_j^s(t) = 0 & t < (1 + 1/\rho)/(\omega_j) \end{cases} \forall j.$$

Prop. 4.5 suggests that when $t$ is large enough, the noise and weak components can be removed in $\theta^s$ and meanwhile, the $\theta^s$, which is the same to the solution in WDI in Eq. 1, can estimate the strong components without bias. On the other hand for the dense parameter $\theta(t)$, it keeps the strong components in $\theta^s$ and meanwhile estimates the weak signals with $\frac{\omega_j}{1+\rho}$ via the $\ell_2$ shrinkage.

To explain, note that starting from $\rho_j(0) = 0$, when $|\rho_j(t)| = 1$ then we have $\dot{\rho}_j(t) = 0$ and hence $\theta_j^s(t) = \theta_j(t)$ according to Eq. 3b, which further has $\theta_j(t) = \omega_j$ according to Eq. 3a. When $|\rho_j(t)| < 1$ for some $j$, then we have $\theta_j^s = 0$ and Eq. 3a gives $\theta_j = \frac{\omega_j}{1+\rho}$. Indeed, the $\theta_j(t)$ for $t > t_j$ given by Eq. 3a is the minimizer of $\frac{1}{2}(\omega_j(t) - \theta_j(t))^2 + \frac{\rho}{2}(\theta_j(t))^2$, where $\frac{\rho}{2}(\theta_j)^2$ can be viewed as an $\ell_2$ regularization of $\theta_j$. Such an $\ell_2$ shrinkage, which is equivalent to the *maximum a posteriori probability* (MAP) estimate with the Gaussian prior $\rho \sim \mathcal{N}(0, 1)$, is similar to the shrinkage effect in the Jame-Stein Estimator which estimates the posterior mean and variance. With such a shrinkage, we show in the following that $(\theta^s(t), \theta(t))$ can estimate the $\theta^{*,s}$ and $\theta^*$ well.

**Theorem 4.6.** *Denote* $\theta_{\max,T}^* := \max_{i \in T} |\theta_i^*|$ *and* $n$ *is large enough such that* $\theta_{\max,T}^* < a_0\sqrt{\log n}$ *for some* $0 < a_0 < 1$. *Then for* $(\theta(t), \theta^s(t))$ *in Eq. 3, if* $n > 4^{1/(1-a_0)}$ *at* $\bar{\tau} := (1 + \frac{1}{\rho})/((1 + a)\sqrt{2\log n})$, *the following holds with probability at least*

$$1 - 2n^{-4a^2} - \max\left(n^{-s/32}, n^{-(1+a)^2/16}\right) - \exp\left(-\sum_{i \in T}(\theta_i^*)^2/72\right) - \exp\left(\frac{-n^{1-a_0}\sum_{i \in T}(\theta_i^*)^2}{24(2 + \log n)}\right)$$

$$- \exp\left(-\frac{|T|}{8}\max\left(1, \sum_{i \in T}(\theta_i^*)^2/|T| - 1\right)\right) - \exp\left(-\frac{|N|}{8}\max\left(1, \sum_{i \in T}(\theta_i^*)^2)/|T| - 1\right)\right):$$

1. **Strong Signal Recovery.** *For the strong signal coefficients* $\theta_S^{*,s}$,

$$\|\theta^s(\bar{\tau}) - \theta^{*,S}\|_2 = \|\theta_S(\bar{\tau}) - \theta_S^{*,s}\|_2 < \|\eta(\omega, \lambda) - \theta^{*,s}\|_2, \forall \lambda > 0. \tag{4}$$

2. **Weak Signal Recovery.** *For the weak signal coefficients* $\theta_T^*$, *there exists* $\infty > \rho^* > 0$ *such that*

$$\|\theta(\bar{\tau})_T - \theta_T^*\|_2 < \|0 - \theta_T^*\|_2 = \|\theta_T^*\|_2 \ i.e., \ \rho = \infty \ Shrinkage \ to \ 0;$$

$$\|\theta(\bar{\tau})_T - \theta_T^*\|_2 < \|\omega_T - \theta_T^*\|_2 = \|\varepsilon_T\|_2 \ i.e., \ \rho = 0 \ No \ Shrinkage.$$

*This bound similarly holds for* $\theta_{S^c}^*$, *where* $S^c := T \cup N$ *contains the weak and null components.*

3. **Whole Signal Recovery.** *For* $\theta^*$, *under the same* $\rho^*$ *in item 2, we have*

$$\|\theta(\bar{\tau}) - \theta^*\|_2 < \|\eta(\omega, \lambda) - \theta^*\|_2, \forall \lambda \geq \sqrt{\log n}.$$

Item 1 inherits the property in Thm. 4.3 for WDI in Eq. 1. Item 2 means we can better estimate the weak components $\theta_T^*$ (and also the noise and weak components together, *i.e.*, $\theta_{T \cup N}^*$) via $\ell_2$

shrinkage. Finally, item 3 means our Splitted WDI is more accurate than the Wavelet Shrinkage method with $\lambda \sim \sqrt{2\log n}$. Although this conclusion may not hold for $\lambda = o(\sqrt{\log n})$, the Wavelet Shrinkage with these $\lambda$'s fails to remove noise in identifying $\theta^{*,s}$.

**Selecting stopping time $\bar{\tau}$ and $\rho^*$.** The $\bar{\tau} := (1 + 1/\rho)/((1 + a)\sqrt{2\log n})$ involves an unknown parameter $a$, which is used to define the level of strong signal coefficients $\theta^{*,s}$ Donoho & Johnstone (1994). Empirically, we can set it to a small constant $1 \geq a \geq 0$ so as to remove other components and identify as many strong components as possible. For $\rho^*$, we will show in Appx. E that $\rho^* := (\|\varepsilon_{T \cup N}\|_2^2 + \sum_{j \in T} \theta_j^* \varepsilon_j)(\|\theta_T\|_2^2 + \sum_{j \in T} \theta_j^* \varepsilon_j) \sim n/(|T|\mathrm{mean}(\theta_T^*))$. This term is approximate $O(1)$ if we believe that the weak signal is dense enough (*i.e.*, $\frac{|T|}{n} \sim O(1)$), *e.g.*, non-burst activity in the low $\beta$ band should be dense enough to be responsive to the medication.

**Solution path generation via Linearization.** One can generate a solution path according to Prop. 4.5. Here we consider another method to generate $(\theta^s(t), \theta(t))_t$ via linearization proposed in Yin et al. (2008). Specifically, we consider the following differential inclusion:

$$\frac{\dot{\theta}(t)}{\kappa} = -\nabla_\theta \overline{\ell}_\rho(\theta, \theta^s) = \omega - \theta(t) + \rho(\theta^s(t) - \theta(t)), \tag{5a}$$

$$\dot{v}(t) = -\nabla_{\theta^s} \overline{\ell}_\rho(\theta, \theta^s) = \rho(\theta(t) - \theta^s(t)), \tag{5b}$$

$$v(t) \in \partial\left(\|\theta^s(t)\|_1 + \frac{1}{2\kappa}\|\theta^s\|_2^2\right) = \rho(t) + \frac{\theta^s(t)}{\kappa}, \tag{5c}$$

where we introduce an $\ell_2$ norm $\frac{1}{2\kappa}\|\theta^s\|_2^2$ for discretization. We show in Appx. E that Eq. 5 also has a closed form solution $(\theta^{s,\kappa}(t), \theta^\kappa(t))$, which is continuous w.r.t. $\kappa$ and converges to $(\theta^s(t), \theta(t))$ in Eq. 3 as $\kappa \to \infty$. Therefore, Thm. 4.6 still holds in Eq. 5 when $\kappa$ is large enough. Moreover, $\frac{1}{2\kappa}\|\theta^s\|_2^2$ enables to implement gradient descent to generate a discrete solution path:

$$\theta_{k+1} = \theta_k + \kappa\delta\left(\omega - \theta(k) + \rho(\theta^s(k) - \theta(k))\right), \tag{6a}$$

$$v(k+1) = v(k) + \delta\rho(\theta(t) - \theta^s(t)), \tag{6b}$$

$$\theta^s(k+1) = \kappa\eta(v(k+1), 1), \tag{6c}$$

where $\delta$ denotes the step size. We show in Appx. F with sufficiently large $\kappa$, Thm. 4.6 still holds in Eq. 6 as long as $\delta < 1/(\kappa(1 + \rho))$. Note that such a discrete corresponds to a general form where the Wavelet matrix may not be orthogonal *i.e.*, $\overline{\ell}_\rho(\theta, \theta^s) := \frac{1}{2}\|y - W\theta\|_2^2 + \frac{\rho}{2}\|\theta - \theta^2\|_2^2$, in which the closed-form solution may not be obtained.

## 5 NUMERICAL EXPERIMENTS

In this section, we conduct numerical experiments to demonstrate the effectiveness of our method in estimating both strong signals and the whole signal.

**Data Synthesis.** We set $n = 1024$ and the 1-$d$ discrete wavelet transform (DWT) matrix $W \in \mathbb{R}^{n \times n}$ as the Daubechies 6 with level 5, which is orthogonal. For the coefficients $\theta^* \in \mathbb{R}^n$, we respectively set the strong signal set and the weak signal set as $S := \{1, 4, 7, ..., 199\}$ and $T := \{401, 403, ..., n\}$. We set $a = 0.3$ and $\theta_i^* = 2(1+a)\sqrt{2\log(n)}$ if $i \in S$; $= 2$ if $i \in T$; and $= 0$ otherwise. We generate the sequence $f = W\theta^* + \varepsilon$ with $\varepsilon_1, ..., \varepsilon_n \sim_{i.i.d} \mathcal{N}(0, 1)$. We denote $\theta^{*,s}$ as the strong signal vector such that $\theta_i^{*,s} = \theta_i^*$ for $i \in S$ and $= 0$ otherwise. We measure the $\ell_2$ error of $\theta^{*,s}$ and $\theta^*$. To remove the effect of randomness, we repeat 20 times.

**Implementations of Our method and Compared Baselines.** We compare with the following threshold value methods that estimate $\hat{\theta} := \eta(W'f, \lambda)$, which includes **i) SURE** Donoho & Johnstone (1995) that selects $\lambda$ based on Stein's Unbiased Risk Estimate; **ii) Universal** method that constantly set $\lambda = \sqrt{2\log(n)}$ Donoho & Johnstone (1994); **iii) Mixture** method that Verma & Verma (2012) combines **SURE** and **Universal**, depending on the signal-to-noise (SNR) ratio. Specifically, if the SNR is high, the **Universal** adopts the same threshold value with the **Universal** method; and **iv) Minimax** Verma & Verma (2012) that selects $\lambda$ using a minimax rule, *i.e.*, $\lambda = (0.3936 + 0.10829 \log_2 n)$ if $n > 32$ and $= 0$ otherwise. For our method, we set $\kappa = 1000$, $\delta = \frac{1}{\kappa(1+\rho)}$, $\rho = \frac{1}{2}$ and the stopping time $\hat{t} = \frac{1+1/\rho}{2\sqrt{2\log n}}$ s.t. our final estimations are $\theta(\hat{t}), \theta^s(\hat{t})$.

Table 1: Avenge ± Std of relative $\ell_2$-error of estimating $\theta^*$ and $\theta^{*,s}$.

|  | SURE | Universal ($\sqrt{2\log(n)}$) | Mixture | Minimax | **Ours** ($\theta(\hat{t})$) | **Ours** ($\theta^s(\hat{t})$) |
|---|---|---|---|---|---|---|
| $\frac{\|\hat{\theta}-\theta^{*,s}\|_2}{\|\theta^{*,s}\|_2}$ | $0.4195 \pm 0.0180$ | $0.3991 \pm 0.0099$ | $0.3991 \pm 0.0099$ | $0.2849 \pm 0.0099$ | $0.4686 \pm 0.0114$ | $\mathbf{0.1400 \pm 0.0373}$ |
| $\frac{\|\hat{\theta}-\theta^*\|_2}{\|\theta^*\|_2}$ | $0.3001 \pm 0.0073$ | $0.5437 \pm 0.0060$ | $0.5437 \pm 0.0060$ | $0.4297 \pm 0.0058$ | $\mathbf{0.2918 \pm 0.0063}$ | $0.3742 \pm 0.0064$ |

**Results Analysis.** We report the average of relative $\ell_2$ error of both $\theta^{*,s}$ and $\theta^*$ in Tab. 1. As shown, our method has a smaller error compared to others. Specifically, for the strong signal $\theta^{*,s}$, all methods except SURE adopt an overly large threshold value ($O \sim \log n$) which can induce errors in estimating strong signals. The SURE method with a smaller threshold value however induces noise components into the estimation, which can explain why SURE also suffers from a large error. For $\theta^*$, our method is comparable to the SURE and outperforms others which drop the weak signals.

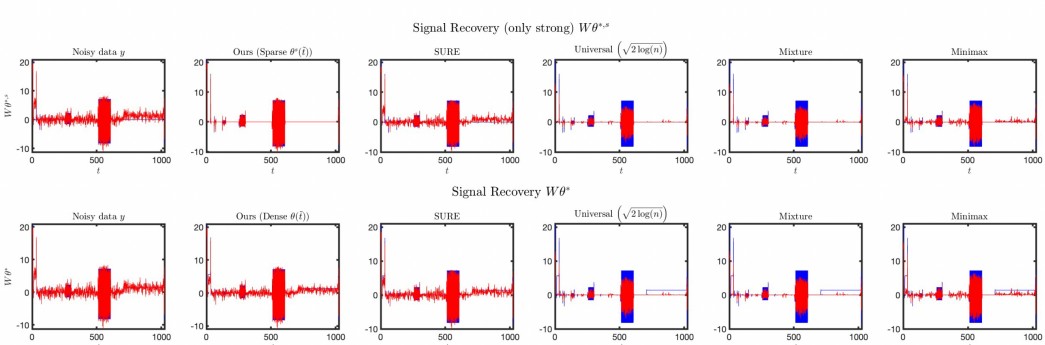

Figure 1: Visualization of Signal Recovery of $W * \theta^{*,s}$ (top) and $W * \theta^*$ (bottom). The blue curve represents the original signal, the red curve represents the estimated one.

**Visualization of Reconstructed Signals.** As shown in Fig. 1, our method can well recover $W\theta^{*,s}$ (top row) and $W\theta^*$ (bottom row); as a contrast, SURE induces additional errors accounted by weak signals and noise in estimating $\theta^{*,s}$ while other methods with excessive shrinkage strategy will suffer from inaccurate estimations of $\theta^{*,s}$ and the ignorance of the weak signals in estimating $\theta^*$.

$\ell_2$ **error along the solution path.** As shown in Fig. 2, the $\ell_2$ error of $\|\theta(t) - \theta^*\|_2$ (blue curve in the right) and $\|\theta^s(t) - \theta^{*,s}\|_2$ (red curve in the left) first decreases then increases as $t$ grows. For $\theta^{*,s}$, the $\theta^s(t)$ continuously identifies more signals until all strong signals are picked up. Meanwhile, the dense parameter $\theta(t)$ can additionally learn weak signals, therefore has a smaller error in estimating $\theta^*$. If $t$ continues to increase, $\theta^s(t)$ will learn weak signals and finally both $\theta^s(t)$ and $\theta(t)$ converge to the noisy coefficients $W'y$. Moreover, our estimated stopping time $\hat{t}$ (blue vertical line) yields a comparable estimation error to the minimum in the solution path (red vertical line).

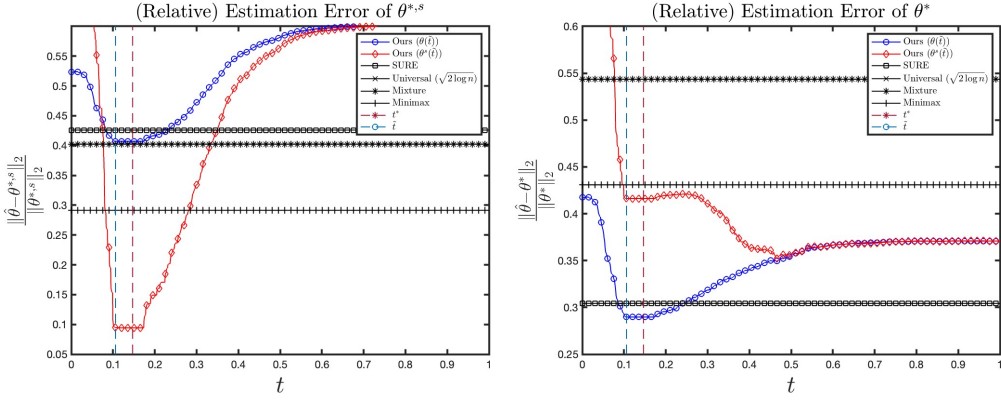

Figure 2: $\ell_2$ error of $\theta^{*,s}$ (left) and $\theta^*$ (right) along the path. The blue (*resp.* red) curve represents the MSE of $\theta(t)$ (*resp.* $\tilde{\theta}(t)$). The blue (*resp.* red) vertical represents the estimated (*resp.* ground-truth) stopping time $\hat{t}$ (*resp.*, $t^*$).

# 6  APPLICATION TO NEURAL SIGNAL RECOGNITION IN PARKINSONIANS

We apply our method to the signal neural signals reconstruction in Parkinson's Disease[1], which can be further used to predict the upcoming movements and study the medication study.

**Data & Problem Description.** We consider the Local field potential signals (LFPs) in the subthalamic nucleus (STN), which are pathologically related Maling & McIntyre (2016); Kühn et al. (2006); Little et al. (2013). For such signals, the $\theta^{*,s}$ corresponds to the burst activity that contains all local synaptic activity from clusters of hundreds of neurons Buzsáki et al. (2012), while $\theta^*$ contains both burst and tonic/non-burst activities. Although increased burst activity of $\beta$ band signals (especially low $\beta$ band Khawaldeh et al. (2022)) has been the most typical biomarker of Parkinson's Disease Lofredi et al. (2023); Tinkhauser et al. (2017), the non-burst signal can further help predict the upcoming movement Khawaldeh et al. (2020).

The data collected in Nie et al. (2021); Wiest et al. (2023)) contains 17 advanced Parkinson's disease patients (34 hemispheres) recruited at St. George's University Hospital NHS Foundation Trust, London, King's College Hospital NHS Foundation Trust, London, and the University Medical Center Mainz. For each patient, we record the resting-state LFPs in the on-medication and off-medication (at least 12h off) states 3 to 7 days after electrode implantation surgery. We adopt Nie et al. (2021) to process these signals. Obvious artifacts (including large baseline fluctuations and muscle activity) were ruled out by visual inspection. For each recording, at least 100s of the raw signal was preserved during processing. Then, low pass filter at 90HZ and high pass filter at 2HZ, and resample to 320Hz. Eliminate power frequency interference through a 50 Hz adaptive notch filter.

**Implementation details.** We follow Luo et al. (2018) to perform a 1-$d$ stationary wavelet transform (SWT) on LFPs as the Symlet 8 with level 6. We follow Donoho & Johnstone (1995) to estimate $\sigma$ as $\tilde{\sigma} = \text{Median}(W_j)/0.6745$. For Wavelet shrinkage, we select $\lambda$ according to the minimax rule in Donoho & Johnstone (1994). To well adapt to each layer, we follow Baldazzi et al. (2020) to multiply $\lambda$ with $1/(\ln j + 1)$ for the layer $j$. For SWDI, we set $\kappa = 20$, $\delta = 1/(\kappa(1 + \rho))$ with $\rho = 0.1$ and the stopping time as $\hat{t} = \frac{1+1/\rho}{\tilde{\sigma}/(\ln j+1)}$.

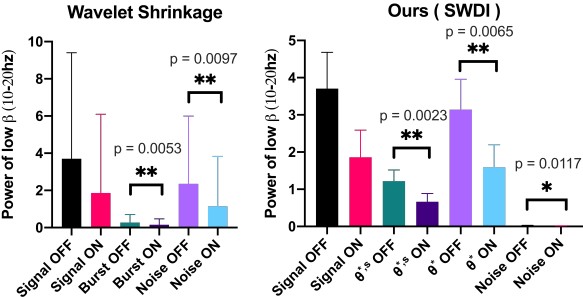

Figure 3: Two-sample T-test on the change of signal's energy after medication.

**Medication Effect.** With reconstructed signals by inverse Wavelet transformation, we implement a two-sample T-test to measure whether the signal's energy is significantly reduced before (*i.e.*, "OFF") and after receiving medications (*i.e.*, "ON"). Here, the energy is defined as the power of low $\beta$, *i.e.*, $E(t) = \frac{1}{W} \sum_{i=t-W+1}^{t} \hat{f}^2(i)$ in Maling & McIntyre (2016), with $W$ denoting the window size. We report the $p$-values in Fig. 3. As shown, the reconstructed strong signal (respectively corresponding to "Burst" in Wavelet Shrinkage and "$\theta^{*,s}$" of our SWDI in Fig. 3) of our method is more significant ($p = 0.0023$) than Wavelet Shrinkage ($p = 0.0053$) in the response to medications, which may due to the effectiveness in bias removal. On the other hand, the significance "noise" component decreases from $p = 0.0097$ to $p = 0.0117$, which is accounted for by the non-burst component that also shows a significant correlation, *i.e.*, $p = 0.0065$.

**Motor Symptoms Improvement.** To measure the effectiveness of the energy's reduction to the symptom relief, we in Fig. 4 report the correlation between the "reduction in the low $\beta$ power" (*i.e.*, energy) and the improvement of motor symptoms measured by the change of clinical Unified

---

[1]We also apply our method to Electroencephalography (EEG) signal denoising, please refer to Appx. H.

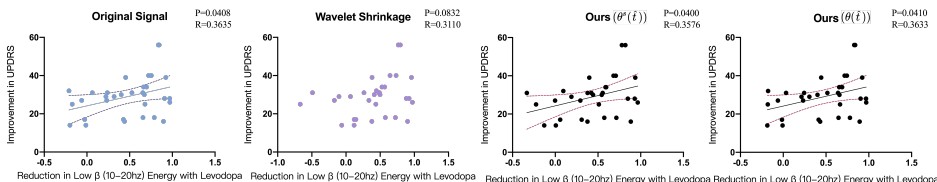

Figure 4: Correlation between changes of signal's energy and the UPDRS that measures the improvement of motor symptoms.

Parkinson's Disease Rating Scale (UPDRS) defined in Goetz et al. (2008). As shown, our reconstructed strong signal $\theta^s$ is significantly correlated to the improvement of motor symptoms while the one given by Wavelet Shrinkage is not. Besides, with additional learned non-burst components which are recognized as noise by Wavelet Shrinkage, the $\theta(\hat{t})$ shows an even stronger correlation.

**Results Analysis.** Fig. 3,. 4 suggest that the reconstructed signals by our method contain more physiological and clinical information. Specifically, the whole signal we learn is not only composed of synchronized/burst activity (in the Low $\beta$ band) that corresponds to high amplitude components; but also other lower components called the non-burst activity. Traditional methods mainly focused on the effectiveness of medication in inhibiting burst activity Lofredi et al. (2023); Tinkhauser et al. (2017); while our method additionally shows that such an inhibition also happens on non-burst activity, which echos a recent finding in Lofredi et al. (2023). To explain how such a non-burst activity affects the LFP signals, a recent study Kajikawa & Schroeder (2011) hypothesized that such a non-burst activity may correspond to the electric field environment of neuron clusters. Since the LFP signal, which appears as a mixture of local potentials from neuron clusters, has been found to be affected by the fluctuation of this field environment Caruso et al. (2018), the change of such a non-burst activity after medications leads to the change of the LFP signal and its energy.

**Signal Recovery.** To further explain the above results, we visualize the recovered coefficients. As shown in Fig. 5, our reconstructed strong signal (marked by blue) can remove the bias; meanwhile the estimated $\theta(t)$ (marked by red) can capture the information of the weak signal.

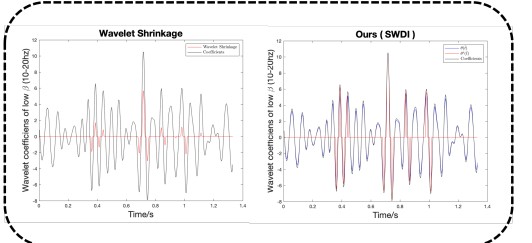 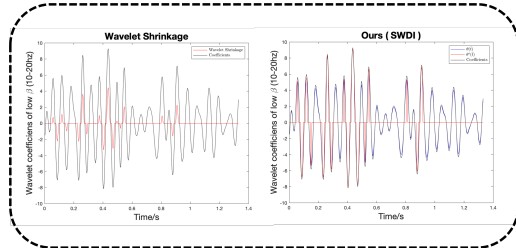

Figure 5: Estimated Signal Coefficients by Wavelet Shrinkage (left) and ours (right).

## 7 CONCLUSION

In this paper, we propose the Splitted Wavelet Differential Inclusion, which can simultaneously remove bias in estimating the strong signal and capture weak components in estimating the whole signal. Our method has a unique closed-form solution. Moreover, we theoretically and empirically show that our method has better estimations than Wavelet Shrinkage. Besides, we provide an efficient discretization algorithm that can efficiently obtain the whole solution path. In Parkinson's Disease analysis, our method identifies the non-burst/tonic activity in the low $\beta$ band, which has been recently found to be responsive to medical treatment.

**Limitations and Future Work.** We only discuss the signal recovery in a non-adaptive way. However, the sub-band and spatially adaptive wavelet decomposition can achieve better reconstruction results. While saying so, our has shown promising results in real applications. Moreover, by considering the strong ad the weak signals on each sub-band, our method can be potentially applied to adaptive decomposition and will be carefully investigated in the future.

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

APPENDIX

## A   RELATED WORK OF DIFFERENTIAL INCLUSION IN SIGNAL RECOVERY

The differential inclusion method in signal recovery was proposed in Osher et al. (2016), which is called (Linearized) *Bregman Inverse Scale Space* and can be viewed as a continuous dynamics of the *Linearized Bregman Iteration* (LBI) proposed in Osher et al. (2005); Yin et al. (2008) for image denoising. From the perspective of differential inclusion, Osher et al. (2016) firstly showed the model selection consistency property, recovering the true signal set under irrepresentable conditions. The Huang & Yao (2018) later generalized this result to from the linear model to a general convex function. Moreover, Huang et al. (2016; 2020) proposed the variable splitting method which leads to better model selection consistency. The Sun et al. (2017) further applied them to Alzheimer's Disease and found there exists another type of lesion feature (which they called "procedural bias") that can help disease diagnosis. Our method is motivated by differences from this method in the Wavelet Denoising scenario, in which our primary goal is to reconstruct the signal. Besides, thanks to the orthogonality of the Wavelet Matrix, our method has a closed-form solution, which leads to theoretical advantages over Wavelet Shrinkage.

## B   SUPPORTING LEMMAS

**Lemma B.1** (Concentration for Lipschitz functions)**.** *Let* $(X_1, \ldots, X_n)$ *be a vector of i.i.d. standard Gaussian variables, and let* $f : \mathbb{R}^n \to \mathbb{R}$ *be L-Lipschitz with respect to the Euclidean norm. Then the variable* $f(X) - \mathbb{E}[f(X)]$ *is sub-Gaussian with parameter at most L, and hence*

$$\mathbb{P}[|f(X) - \mathbb{E}[f(X)]| \geq t] \leq 2e^{-\frac{t^2}{2L^2}} \quad \textit{for all } t \geq 0.$$

**Lemma B.2** (Hoeffding bound)**.** *Suppose that the variables* $X_i, i = 1, \ldots, n$ *are independent, and* $X_i$ *has mean* $\mu_i$ *and sub-Gaussian parameter* $\sigma_i$*. Then for all* $t \geq 0$*, we have*

$$\mathbb{P}\left[\sum_{i=1}^{n}(X_i - \mu_i) \geq t\right] \leq \exp\left\{-\frac{t^2}{2\sum_{i=1}^{n}\sigma_i^2}\right\}.$$

**Lemma B.3** ($\chi^2$-variables)**.** *Let* $(X_1, \ldots, X_n)$ *be a vector of i.i.d. standard Gaussian variables. Then* $(X_1^2, \ldots, X_n^2)$ *are i.i.d Chi-squared variables with* $1$ *degree of freedom. Then we have*

$$\mathbb{P}\left[\frac{1}{n}\left|\sum_{i=1}^{n}X_i^2 - n\right| \geq t\right] \leq e^{-\frac{nt^2}{8}}, \quad \forall t \in (0, 1).$$

**Lemma B.4** (Expectation of Maximum Gaussian (Theorem 1 in Kamath (2015)))**.** *Let Let* $(X_1, \ldots, X_n)$ *be a vector of i.i.d. standard Gaussian variables, then*

$$\mathbb{E}[\max_{1 \leq i \leq n} X_i] \leq \sqrt{2\log n}.$$

$$|f(Ax) - f(Ay)| \leq \sqrt{\max_{i=1,\ldots,n}(A^T A)_{ii}}\|x - y\|_2, \quad x, y \in \mathbb{R}^n.$$

## C   PROOF OF SECTION 3

**Proposition C.1.** *We have* $0 = \arg\inf_\lambda \sup_{|\theta_i^*| \geq 1} \mathbb{E}(\eta(\omega_i, \lambda) - \theta_i^*)^2$.

*Proof.* Since $\theta_i^* \geq 1$, then

$$\arg\inf_\lambda \sup_{\theta_i^* \geq 1} \frac{\mathbb{E}(\eta(\omega_i, \lambda) - \theta_i^*)^2}{n^{-1} + \min\left((\theta_i^*)^2, 1\right)} = \arg\inf_\lambda \sup_{\theta_i^* \geq 1} \mathbb{E}(\eta(\omega_i, \lambda) - \theta_i^*)^2$$

According to Theorem 2 in Donoho & Johnstone (1994), we have

$$\mathbb{E}(\eta(\omega_i, \lambda) - \theta_i^*)^2 = 1 + \lambda^2 + \left((\theta_i^*)^2 - \lambda^2 - 1\right)\{\Phi(\lambda - \theta_i^*) - \Phi(-\lambda - \theta_i^*)\}$$
$$- (\lambda - \theta_i^*)\phi(\lambda + \theta_i^*) - (\lambda + \theta_i^*)\phi(\lambda - \theta_i^*).$$

Denote $g(\lambda, \theta_i*) := \mathbb{E}(\eta(\omega_i, \lambda) - \theta_i^*)^2$. Then it is sufficient to show that for each $\theta_i^*$, we have

$$g(\lambda, \theta_i*) \geq g(\lambda, 0) \geq g(0,0) = g(0, \theta_i*). \tag{7}$$

According to Lemma 1 in Donoho & Johnstone (1994), we have $g(\lambda, \theta_i*)$ is increasing w.r.t. $|\theta_i^*|$, therefore we have $g(\lambda, \theta_i*) \geq g(\lambda, 0)$. The "=" in Eq. 7 is also obvious since

$$g(0, \theta_i*) = 1, \forall\, \theta_i*.$$

It is left to prove the 2nd "$\geq$" in Eq. 7. It is suffcient to show that

$$g(\lambda, 0) = (1 + \lambda^2)(1 - \Phi(\lambda) + \Phi(-\lambda)) - 2\lambda\phi(\lambda)$$

is non-increasing w.r.t. $\lambda$. Take the gradient of $g(\lambda, 0)$ w.r.t. $\lambda$, we have

$$\begin{aligned}
\frac{\partial g(\lambda, 0)}{\partial \lambda} &= -2(1 + \lambda^2)\phi(\lambda) + 2\lambda(1 - \Phi(\lambda) + \Phi(-\lambda)) - 2\phi(\lambda) + 2\lambda^2\phi(\lambda) \\
&= -4\phi(\lambda) + 2\lambda(1 - \Phi(\lambda) + \Phi(-\lambda)).
\end{aligned}$$

Then it is sufficient to show that $\frac{\partial g(\lambda, 0)}{\partial \lambda} \leq 0$. To show this, we first consider the 2nd derivative of $g(\lambda, 0)$, which gives

$$\begin{aligned}
\frac{\partial^2 g(\lambda, 0)}{\partial \lambda^2} &= 4\lambda\phi(\lambda) + 2(1 - \Phi(\lambda) + \Phi(-\lambda)) + 2\lambda(-2\phi(\lambda)) \\
&= 2(1 - \Phi(\lambda) + \Phi(-\lambda)) \geq 0.
\end{aligned}$$

This means $l(\lambda) := \frac{\partial g(\lambda, 0)}{\partial \lambda}$ is non-decreasing. We consider $\lim_{\lambda \to \infty} l(\lambda)$, according to the Hôpital's rule, we have

$$\lim_{\lambda \to \infty} l(\lambda) = \frac{\lim_{\lambda \to \infty} 2(1 - \Phi(\lambda) + \Phi(-\lambda))}{\lim_{\lambda \to \infty} \frac{1}{\lambda}} = \frac{\lim_{\lambda \to \infty} 4\phi(\lambda)}{\lim_{\lambda \to \infty} \frac{1}{\lambda^2}} = \lim_{\lambda \to \infty} 4\lambda^2\phi(\lambda) = 0.$$

This means $l(\lambda) := \frac{\partial g(\lambda, 0)}{\partial \lambda} \leq 0$. Therefore, we have $g(\lambda, 0) \geq g(0, 0)$. Hence,

$$\arg\inf_\lambda \sup_{\theta_i^* \geq 1} \mathbb{E}(\eta(\omega_i, \lambda) - \theta_i^*)^2 = 0.$$

The proof is finished. $\qquad\square$

## D  PROOF OF SECTION 4.1

**Proposition D.1.** *The solution of differential inclusion in Eq. 1 is $\theta_j(t) = \omega_j(t)$ for $t \geq \frac{1}{|\omega_j|}$; and $= 0$ otherwise for each $j$.*

*Proof.* Note that $\rho(0) = \theta(0) = 0$. We define $t_j := \sup_t\{\rho_j(0) + t\omega_j \in \partial|\theta_j(0)|\}$. Then we define $\theta_j(t) = 0$ for $t < t_j$ and $= \omega_j$ for $t \geq t_j$ and $\rho_j(t) = \omega_j t$ for $t < t_j$ and $= \text{sign}(\omega_j)$ for $t \geq t_j$. It can be shown that this defined $\{\rho(t), \theta(t)\}$ is the solution of Eq. 1. This solution is unique since the loss $\ell(\theta(t)) := \frac{1}{2}\|\omega - \theta(t)\|_2^2$ is strictly convex w.r.t. $\theta$. According to Theorem 2.1 in Osher et al. (2016), we know that the $\theta(t)$ is unique. $\qquad\square$

**Theorem D.2.** *Denote $\theta_{\min}^{*,s} := \min_{i \in S}|\theta_i^*|$. Then at $\bar{\tau} := \frac{1}{(1+a)\sqrt{2\log n}}$ for some $a > 0$ and $\theta_{\min}^{*,s} \geq 2(1+a)\sqrt{2\log n}$ and $\frac{a}{2}\sqrt{2\log n} > \theta_j^*$ for $j \notin S$. Then with probability at least $1 - \frac{2}{n^{4a^2}} - \max\left(\exp\left(-\frac{s\lambda^2}{8}\right), \frac{1}{n^{\frac{(1+a)^2}{4}}}\right)$, we have*

$$\|\theta(\bar{\tau}) - \theta^{*,s}\|_2 < \|\eta(\omega, \lambda) - \theta^{*,s}\|_2, \tag{8}$$

*where $\theta(t)$ is the solution of Eq. 1.*

*Proof.* First we show the model selection consistency: $\text{supp}(\theta(\bar{\tau})) = S$. According to Prop. D.1, it is sufficient to show that $\bar{\tau} \geq \frac{1}{|\omega_j := \theta_j^{*,s} + \varepsilon_j|}$ for all $j \in S$ and $\bar{\tau} < \frac{1}{|\omega_j := \theta_j^* + \varepsilon_j|}$ for all $j \notin S$. Since $|\theta_j^*| \leq \frac{a}{2}\sqrt{2\log n}$, it is sufficient to show that $\max_{1 \leq j \leq n}|\varepsilon_j| \leq (1 + \frac{a}{2})\sqrt{2\log n}$ with high

probability, which can ensure that $|\theta_j^{*,s} + \varepsilon_j| \geq \theta_{\min}^{*,s} - \max_{1 \leq j \leq n} |\varepsilon_j| > (1+a)\sqrt{2\log n} = \frac{1}{\bar{\tau}}$ for $j \in S$ and $|\theta_j^* + \varepsilon_j| < \frac{a}{2}\sqrt{2\log n} + \max_{1 \leq j \leq n} |\varepsilon_j| \leq (1+a)\sqrt{2\log n} = \frac{1}{\bar{\tau}}$. Since $\max_{1 \leq j \leq n} |\varepsilon_j| < \max\{\max_{1 \leq n} \varepsilon_j, \max_{1 \leq n} -\varepsilon_j\}$, then we have

$$P(\max_{1 \leq j \leq n} |\varepsilon_j| > (1 + \frac{a}{2})\sqrt{2\log n})$$

$$\leq P\left(\max_{1 \leq j \leq n} \varepsilon_j > (1 + \frac{a}{2})\sqrt{2\log n}\right) + P\left(\max_{1 \leq j \leq n} -\varepsilon_j > (1 + \frac{a}{2})\sqrt{2\log n}\right)$$

$$\leq 2P\left(\max_{1 \leq j \leq n} \varepsilon_j > (1 + \frac{a}{2})\sqrt{2\log n}\right) = 2P\left(\max_{1 \leq j \leq n} \varepsilon_j - \sqrt{2\log n} > \frac{a}{2}\sqrt{2\log n}\right)$$

$$\leq 2P\left(\max_{1 \leq j \leq n} \varepsilon_j - \mathbb{E}\left[\max_{1 \leq j \leq n} \varepsilon_j\right] > \frac{a}{2}\sqrt{2\log n}\right).$$

According to lemma **??** and lemma B.1, we have

$$P\left(\max_{1 \leq j \leq n} \varepsilon_j - \mathbb{E}\left[\max_{1 \leq j \leq n} \varepsilon_j\right] > \frac{a}{2}\sqrt{2\log n})\right) < \frac{1}{n^{\frac{a^2}{4}}}.$$

To prove Eq. 25, without loss of generality we assume that $\theta_j^{*,s} > 0$ for $j \in S$. Then we have

$$\|\theta(\bar{\tau}) - \theta^{*,s}\|_2^2 = \|\varepsilon_S\|_2^2 \tag{9}$$

and that

$$\left(\eta(\omega, \lambda)_j - \theta_j^{*,s}\right)^2 = \begin{cases} (\theta_j^{*,s})^2, & \lambda \geq \theta_j^{*,s} + \varepsilon_j \\ (\varepsilon_j - \lambda)^2, & \lambda < \theta_j^{*,s} + \varepsilon_j \end{cases}$$

We denote $\mathrm{I} := \{j : \lambda \geq \theta_j^{*,s} + \varepsilon_j\}$ and $\mathrm{II} = S - \mathrm{I}$. Then we have

$$\|\eta(\omega, \lambda) - \theta^{*,s}\|_2^2 \geq \|\eta(\omega, \lambda)_S - \theta^{*,s}\|_2^2 = \|\theta_\mathrm{I}^{*,s}\|_2^2 + \|\varepsilon_\mathrm{II}\|_2^2 - 2\lambda \sum_{j \in \mathrm{II}} \varepsilon_j + \lambda^2(s - |\mathrm{I}|) \tag{10}$$

If $\lambda < (1+a)\sqrt{2\log n}$, then condition on $\max_{1 \leq j \leq n} |\varepsilon_j| \leq (1 + \frac{a}{2})\sqrt{2\log n}$, we have $\mathrm{I} = \emptyset$. Combining Eq. 9 and Eq. 10, it is sufficient to show that

$$\sum_{j \in S} \varepsilon_j < \frac{s\lambda}{2},$$

which has probability at least $1 - \exp\left(\frac{-s\lambda^2}{8}\right)$ applying Lemma B.2. Otherwise, if $\lambda \geq (1+a)\sqrt{2\log n}$, then condition on $\max_{1 \leq j \leq n} |\varepsilon_j| \leq (1 + \frac{a}{2})\sqrt{2\log n}$, we have $\|\varepsilon_\mathrm{I}\|_2^2 < \|\theta_\mathrm{I}^{*,s}\|_2^2$. Combining Eq. 9 and Eq. 10, it is sufficient to show that

$$\sum_{j \in \mathrm{II}} \varepsilon_j < \frac{|\mathrm{II}|\lambda}{2},$$

which has probability at least $1 - \frac{1}{n^{\frac{(1+a)^2}{4}}}$ applying Lemma B.2. The proof is completed. $\square$

## D.1  Linearized Wavelet Differential Inclusion

Similar to SWDI, we also provide a linearized version in the following:

$$\dot{\rho}(t) + \frac{\dot{\theta}(t)}{\kappa} = \omega - \theta(t), \tag{11a}$$

$$\rho(t) \in \partial\|\theta(t)\|_1. \tag{11b}$$

**Proposition D.3.** *The solution of differential inclusion in Eq. 11 is*

$$\theta_j(t) = \begin{cases} \omega_j \left(1 - \exp\left(-\kappa\left(t - \frac{1}{\omega_j}\right)\right)\right), & t \geq \frac{1}{|\omega_j|} \\ 0, & otherwise \end{cases} \quad \forall j.$$

*Proof.* Similar to Prop. D.1, we define $t_j$ for each $j$ and $\rho_j(t) = \omega_j t$ for $t < t_j$ and $= \text{sign}(\theta_j)$ for $t \geq t_j$. It is easy to validate that such defined $(\rho(t), \theta(t))$ is the solution of Eq. 11. To see the uniqueness, we denote $v(t) := \rho(t) + \frac{\theta(t)}{\kappa}$, then we have

$$\dot{z}(t) = \omega - \kappa\eta(z(t), 1) := g(z(t)).$$

Since $g(\cdot)$ is Lipschitz continuous, the solution is unique according to the Picard-Lindelöf Theorem.
$\square$

**Theorem D.4.** *Under the same conditions and the definition of $\bar{\tau}$ in Thm. D.2. Then with probability at least* $1 - \frac{2}{n^{4a^2}} - \max\left(\exp\left(-\frac{s\lambda^2}{32}\right), \frac{1}{n^{\frac{(1+a)^2}{16}}}\right)$, *we have*

$$\|\theta(\bar{\tau}) - \theta^{*,s}\|_2 < \|\eta(\omega, \lambda) - \theta^{*,s}\|_2, \tag{12}$$

*where $\theta(t)$ is the solution of Eq. 11.*

*Proof.* Similar to Thm. D.2, conditioning on $\max_{1 \leq j \leq n} |\varepsilon_j| \leq (1 + \frac{a}{2})\sqrt{2\log n}$ (with probability at least $1 - \frac{2}{n^{4a^2}}$), we have that $\text{supp}(\theta(\bar{\tau})) = S$, by additionally noting that $\bar{\tau}$ is exactly greater than $\frac{1}{\omega_j}$ for any $j \in S$. Next, we show that with probability at least $1 - \max\left(\exp\left(-\frac{s\lambda^2}{32}\right), \frac{1}{n^{\frac{(1+a)^2}{16}}}\right)$, we have

$$\|\bar{\theta}(\bar{\tau}) - \theta^{*,s}\|_2^2 < \|\eta(\omega, \lambda) - \theta^{*,s}\|_2^2 - \min\left(\frac{\lambda^2}{2}, (1+a)^2 \log n\right), \tag{13}$$

where $\bar{\theta}(t) = \lim_{\kappa \to \infty} \theta_\kappa(t)$ with $\theta_\kappa(t)$ being the solution of Eq. 11 with a fixed $\kappa > 0$. According to Prop. D.1, D.3, the $\bar{\theta}(t)$ is the solution of Eq. 11. If Eq. 13 holds, then due to the continuity of $\theta_\kappa(t)$ with respect to $\kappa$, we have Eq. 12 as long as $\kappa$ is large enough. Similar to the proof for Thm. D.2, when $\lambda < (1+a)\sqrt{2\log n}$, conditioning on $\max_{1 \leq j \leq n} |\varepsilon_j| \leq (1 + \frac{a}{2})\sqrt{2\log n}$ we have $\sum_{j \in S} \varepsilon_j < \frac{s\lambda}{4}$ and hence

$$\|\bar{\theta}(\bar{\tau}) - \theta^{*,s}\|_2^2 < \|\eta(\omega, \lambda) - \theta^{*,s}\|_2^2 - \frac{\lambda^2}{2}$$

with probability at least $1 - \exp\left(\frac{-s\lambda^2}{32}\right)$. When $\lambda \geq (1+a)\sqrt{2\log n}$, we have $\sum_{j \in \text{II}} \varepsilon_j < \frac{|\text{II}|\lambda}{2}$ and therefore

$$\|\bar{\theta}(\bar{\tau}) - \theta^{*,s}\|_2^2 < \|\eta(\omega, \lambda) - \theta^{*,s}\|_2^2 - (1+a)^2 \log n$$

with probability at least $1 - \frac{1}{n^{\frac{(1+a)^2}{16}}}$.
$\square$

## E  PROOF OF SECTION 4.2

**Proposition E.1.** *The solution of differential inclusion in Eq. 3 is*

$$\begin{cases} \theta_j(t) = \gamma_j(t) = \omega_j, & t \geq \frac{1+\frac{1}{\beta}}{\omega_j} \\ \theta_j(t) = \frac{\omega_j}{1+\beta}, \ \gamma_j(t) = 0 & t < \frac{1+\frac{1}{\beta}}{\omega_j} \end{cases} \forall j.$$

*Proof.* Note that $\gamma(0) = \rho(0) = 0$. Then we define $t_j$ for each $j$ as

$$t_j := \sup_{t>0}\{\rho_j(0) + \frac{t\omega_j}{1+\beta} \notin \partial|\gamma_j(0)|\}.$$

When $t < t_j$, we have $|\rho_j(t)| < 1$ thus $\gamma_j(t) = 0$ and also $\theta_j = \frac{\omega_j}{1+\beta}$ according to Eq. 3a. For $t \geq t_j$, we have $|\rho_j(t)| = 1$ and thus $\dot{\rho}_j(t) = 0$. Therefore, we have $\gamma_j(t) = \theta_j(t) = \omega_j$. It is easy to see such defined $(\rho(t), \theta(t), \gamma(t))$ is the solution of Eq. 3. We can obtain that $t_j = \frac{1+\frac{1}{\beta}}{\omega_j}$. According to Huang et al. (2016), this solution is unique.
$\square$

**Theorem E.2.** *Denote $\theta^*_{\max,T} := \max_{i \in T} |\theta^*_i|$ and $n$ is large enough such that $\theta^*_{\max,T} < a_0\sqrt{\log n}$ for some $0 < a_0 < 1$. Then for $(\theta(t), \theta^s(t))$ in Eq. 3, if $n > 4^{1/(1-a_0)}$ at $\bar{\tau} := \frac{1+\frac{1}{\rho}}{(1+a)\sqrt{2\log n}}$, the following holds with probability at least*

$$1 - \frac{2}{n^{4a^2}} - \max\left(\frac{1}{n^{\frac{s}{32}}}, \frac{1}{n^{\frac{(1+a)^2}{16}}}\right) - \exp\left(-\frac{\sum_{i \in T}(\theta^*_i)^2}{72}\right) - \exp\left(\frac{-n^{1-a_0}\sum_{i \in T}(\theta^*_i)^2}{24(2+\log n)}\right)$$

$$- \exp\left(-\frac{|T|\max\left(1, \frac{\sum_{i \in T}(\theta^*_i)^2}{|T|} - 1\right)}{8}\right) - \exp\left(-\frac{|N|\max\left(1, \frac{\sum_{i \in T}(\theta^*_i)^2}{|T|} - 1\right)}{8}\right):$$

1. **Strong Signal Recovery.** *For the strong signal coefficients $\theta^{*,s}_S$,*

$$\|\theta^s(\bar{\tau}) - \theta^{*,S}\|_2 = \|\theta_S(\bar{\tau}) - \theta^{*,s}_S\|_2 < \|\eta(\omega, \lambda) - \theta^{*,s}\|_2, \ \forall \lambda > 0. \tag{14}$$

2. **Weak Signal Recovery.** *For the weak signal coefficients $\theta^*_T$, there exists $\infty > \rho^* > 0$ such that*

$$\|\theta(\bar{\tau})_T - \theta^*_T\|_2 < \|0 - \theta^*_T\|_2 = \|\theta^*_T\|_2 \ i.e., \ \rho = \infty \ Shrinkage \ to \ 0;$$
$$\|\theta(\bar{\tau})_T - \theta^*_T\|_2 < \|\omega_T - \theta^*_T\|_2 = \|\varepsilon_T\|_2 \ i.e., \ \rho = 0 \ No \ Shrinkage.$$

*We can obtain a similar result for $\theta^*_{S^c}$, where $S^c := T \cup N$ contains the weak and null components.*

3. **Whole Signal Recovery.** *For $\theta^*$, under the same $\rho^*$ in item 2, we have*

$$\|\theta(\bar{\tau}) - \theta^*\|_2 < \|\eta(\omega, \lambda) - \theta^*\|_2, \ \forall \lambda \geq \sqrt{\log n}.$$

*Proof.* The proof of **Strong Signal Recovery** is the same as the proof for Thm. D.2. For **Weak Signal Recovery**, we define $I(p) = \|(1-p)\omega_T - \theta^*_T\|^2_2 = \|p\theta^*_T - (1-p)\varepsilon_T\|^2_2$. We then have

$$I'(p) = 2p\sum_{i \in T}(\theta^*_i + \varepsilon_i)^2 - 2\sum_{i \in T}\varepsilon_i(\theta^*_i + \varepsilon_i).$$

We obtain the minimizer of $I(p)$ as $p^* = \frac{\sum_{i \in T}\varepsilon_i^2 + \sum_{i \in T}\varepsilon_i\theta^*_i}{\sum_{i \in T}(\theta^*_i + \varepsilon_i)^2}$ by setting $I(p') = 0$. If we can show that $0 < p^* < 1$, then there exists a $\beta^*$ such that $p^* = \frac{\beta^*}{1+\beta^*}$. Then it is sufficient to show that

$$\sum_i \varepsilon_i\theta^*_i + \sum_i \varepsilon_i^2 > 0, \tag{15}$$

$$\sum_i \varepsilon_i\theta^*_i + \frac{1}{6}\sum_i \theta^*_i > 0. \tag{16}$$

For Eq. 15, we have that:

$$P\left(\sum_i \varepsilon_i\theta^*_i + \sum_i \varepsilon_i^2 < 0\right)$$

$$= P\left(\sum_i \varepsilon_i\theta^*_i + \sum_i \varepsilon_i^2 < 0, \sum_i \varepsilon_i^2 < \frac{|T|}{2}\right) + P\left(\sum_i \varepsilon_i\theta^*_i + \sum_i \varepsilon_i^2 < 0, \sum_i \varepsilon_i^2 \geq \frac{|T|}{2}\right)$$

$$\leq P\left(\sum_i \varepsilon_i^2 < \frac{|T|}{2}\right) + P\left(\sum_i \varepsilon_i\theta^*_i < -\frac{|T|}{2}\right)$$

Applying Lemma B.3 to the first term and Lemma B.2 to the second term, we have that

$$P\left(\sum_i \varepsilon_i^2 < \frac{|T|}{2}\right) < \exp\left(-\frac{|T|}{8}\right), \ P\left(\sum_i \varepsilon_i\theta^*_i < -\frac{|T|}{2}\right) < \exp\left(-\frac{|T|^2}{8\sum_{i \in T}(\theta^*_i)^2}\right).$$

For Eq. 16, we have

$$P\left(\sum_i \varepsilon_i\theta^*_i + \frac{1}{4}\sum_i(\theta^*_i)^2 < 0\right) = P\left(\sum_i \varepsilon_i\theta^*_i < -\frac{\sum_i(\theta^*_i)^2}{6}\right).$$

Applying Lemma B.2, we have that

$$P\left(\sum_i \varepsilon_i \theta_i^* < -\frac{\sum_{i \in T}(\theta_i^*)^2}{4}\right) \leq \exp\left(-\frac{\sum_{i \in T}(\theta_i^*)^2}{72}\right).$$

Since $\|\theta_T^*\|_2$ is the $\ell_2$ error of $0_T$ with $p = 1$ and $\|\varepsilon_T\|_2$ is the $\ell_2$ error of $\omega_T$ with $p = 0$, we obtain the conclusion. To extend this result to $\theta_{S^c}^*$, following the same procedure, we can obtain that

$$p^* = \frac{\sum_{i \in T}\varepsilon_i^2 + \sum_{i \in T}\varepsilon_i \theta_i^* + \sum_{j \in N}\varepsilon_j^2}{\sum_{i \in T}(\theta_i^* + \varepsilon_i)^2 + \sum_{j \in N}\varepsilon_j^2},$$

which is $< 1$ and $> 0$ if Eq. 15, 16 holds. Finally, we prove **the Whole Signal Recovery**. According to the above results, it is sufficient to show that there exists a $\beta$ such that

$$\|\theta(\bar{\tau})_{S^c} - \theta_{S^c}^*\|_2 < \|\eta(\omega, \lambda)_T - \theta_T^*\|_2 < \|\eta(\omega, \lambda)_{S^c} - \theta_{S^c}^*\|_2.$$

condition on $\|\theta(\bar{\tau})_S - \theta_S^*\|_2 < \|\eta(\omega, \lambda)_S - \theta_S^*\|_2$ that holds with high probability. We first show that

$$\|\eta(\omega, \lambda)_T - \theta_T^*\|_2^2 > \frac{1}{3}\|\theta_T^*\|_2^2, \tag{17}$$

$$\|\theta(\bar{\tau})_{S^c} - \theta_{S^c}^*\|_2^2 < \frac{2}{3}\|\theta_T^*\|_2^2, \tag{18}$$

for some $\infty > \beta > 0$. For Eq. 17, we denote $b_i := \eta(\omega, \lambda)_i$, then we have

$$\|\eta(\omega, \lambda)_T - \theta_T^*\|_2^2 = \sum_{i \in T} b_i^2 + \sum_{i \in T} 2\theta_i^* b_i + \sum_{i \in T}(\theta_i^*)^2 \geq \sum_{i \in T} 2\theta_i^* b_i + \sum_{i \in T}(\theta_i^*)^2,$$

which holds as long as $\sum_{i \in T}\theta_i^* b_i \geq -\frac{1}{6}\sum_{i \in T}(\theta_i^*)^2$. We then have

$$P\left(\sum_{i \in T}\theta_i^* b_i \leq -\frac{1}{6}\sum_{i \in T}(\theta_i^*)^2\right) = P\left(\sum_{i \in T}\theta_i^* b_i - \sum_{i \in T}\theta_i^*\mathbb{E}[b_i] \leq -\frac{1}{6}\sum_{i \in T}(\theta_i^*)^2 - \sum_{i \in T}\theta_i^*\mathbb{E}[b_i]\right)$$

$$\leq \exp\left(-\frac{\left(\frac{1}{6}\sum_{i \in T}(\theta_i^*)^2 + \sum_{i \in T}\theta_i^*\mathbb{E}[b_i]\right)^2}{2\sum_{i \in T}(\theta_i^*)^2\mathbb{E}[b_i^2]}\right). \tag{19}$$

Without loss of generality, we assume $\theta_i^* > 0$. Then for $\mathbb{E}[b_i]$, we have

$$\mathbb{E}[b_i] = \frac{1}{\sqrt{2\pi}}\left(\int_{\sqrt{\log n}-\theta_i^*}^{+\infty} x\exp\left(-\frac{x^2}{2}\right)dx + \int_{-\infty}^{-\sqrt{\log n}-\theta_i^*} x\exp\left(-\frac{x^2}{2}\right)dx\right)$$

$$= -\frac{1}{\sqrt{2\pi}}\int_{-\sqrt{\log n}-\theta_i^*}^{-\sqrt{\log n}+\theta_i^*} x\exp\left(-\frac{x^2}{2}\right)dx$$

$$> -\frac{1}{\sqrt{2\pi}}\frac{1}{n^{1-\frac{\theta_i^*}{\sqrt{\log n}}}} \geq -\frac{1}{\sqrt{2\pi}}\frac{1}{n^{1-a_0}}.$$

For $\mathbb{E}[b_i^2]$, we have

$$\mathbb{E}[b_i^2] = \frac{1}{\sqrt{2\pi}}\left(\underbrace{\int_{\sqrt{\log n}-\theta_i^*}^{+\infty} x\exp\left(-\frac{x^2}{2}\right)dx}_{J_1} + \underbrace{\int_{-\infty}^{-\sqrt{\log n}-\theta_i^*} x\exp\left(-\frac{x^2}{2}\right)dx}_{J_2}\right).$$

For $J_1$, we have

$$\frac{1}{\sqrt{2\pi}}J_1 = \frac{1}{\sqrt{2\pi}}\int_{\sqrt{\log n}-\theta_i^*}^{+\infty} xd\left(-\exp\left(-\frac{x^2}{2}\right)\right)$$

$$= \frac{1}{\sqrt{2\pi}} - x\exp\left(-\frac{x^2}{2}\right)\Big|_{\sqrt{\log n}-\theta_i^*}^{+\infty} + P(\varepsilon_i > \sqrt{\log n} - \theta_i^*)$$

$$= P(\varepsilon_i > \sqrt{\log n} - \theta_i^*) + \frac{1}{\sqrt{2\pi}}\left(\sqrt{\log n} - \theta_i^*\right)\exp\left(-\frac{\left(\sqrt{\log n} - \theta_i^*\right)^2}{2}\right)$$

Similarly, for $J_2$, we have

$$\frac{1}{\sqrt{2\pi}}J_2 = P(\varepsilon_i > \sqrt{\log n} + \theta_i^*) + \frac{1}{\sqrt{2\pi}}\left(\sqrt{\log n} + \theta_i^*\right)\exp\left(-\frac{\left(\sqrt{\log n} + \theta_i^*\right)^2}{2}\right)$$

$$\leq P(\varepsilon_i > \sqrt{\log n} - \theta_i^*) + \frac{1}{\sqrt{2\pi}}\left(\sqrt{\log n} + \theta_i^*\right)\exp\left(-\frac{\left(\sqrt{\log n} - \theta_i^*\right)^2}{2}\right).$$

Therefore, we have

$$\mathbb{E}[b_i^2] \leq 2P(\varepsilon_i > \sqrt{\log n} - \theta_i^*) + \frac{2}{\sqrt{2\pi}}\sqrt{\log n}\exp\left(-\frac{\left(\sqrt{\log n} - \theta_i^*\right)^2}{2}\right) \leq \frac{2 + \log n}{n^{1-a_0}}.$$

Substituting these results into Eq. 19, we have:

$$P\left(\sum_{i \in T}\theta_i^* b_i \leq -\frac{1}{6}\sum_{i \in T}(\theta_i^*)^2\right) \leq \exp\left(\frac{-n^{1-a_0}\left(\frac{1}{6} - \frac{1}{\sqrt{2\pi}n^{1-a_0}}\right)\sum_{i \in T}(\theta_i^*)^2}{2(2 + \log n)}\right),$$

$$\leq \exp\left(\frac{-n^{1-a_0}\sum_{i \in T}(\theta_i^*)^2}{24(2 + \log n)}\right),$$

as long as $n > 4^{1/(1-a_0)}$. Next we prove Eq. 18, which is equivalent to showing that

$$\sum_{i \in T}\varepsilon_i^2 + \sum_{j \in N}\varepsilon_j^2 - 2\beta\sum_{i \in T}\theta_i^*\varepsilon_i < \frac{2(2\beta + 1)}{3}\sum_{i \in T}(\theta_i^*)^2.$$

If we take $\beta = n/|T|$, then it is sufficient to show that

$$\frac{\sum_{i \in T}\varepsilon_i^2}{|T|} \leq \frac{1}{|T|}\sum_{i \in T}(\theta_i^*)^2, \tag{20}$$

$$\frac{\sum_{j \in N}\varepsilon_j^2}{|N|} \leq \frac{1}{|T|}\sum_{j \in T}(\theta_j^*)^2, \tag{21}$$

$$\sum_{i \in T}\varepsilon_i\theta_i^* + \frac{1}{6}\sum_{j \in T}(\theta_i^*)^2 > 0. \tag{22}$$

Conditioning on $\sum_i \varepsilon_i\theta_i^* + \frac{1}{6}\sum_i(\theta_i^*)^2 > 0$, $\sum_{i \in T}\varepsilon_i^2 \geq \frac{|T|}{2}$ and $\|\theta(\bar{\tau})_S - \theta_S^*\|_2 < \|\eta(\omega, \lambda)_S - \theta_S^*\|_2$, we have that Eq. 22 hold. For Eq. 20, we have

$$P\left(\frac{\sum_{i \in T}\varepsilon_i^2}{|T|} > \frac{1}{|T|}\sum_{i \in T}(\theta_i^*)^2\right) \leq \exp\left(-\frac{|T|\max\left(1, \frac{\sum_{i \in T}(\theta_i^*)^2}{|T|} - 1\right)}{8}\right).$$

Similarly, for Eq. 21, we have

$$P\left(\frac{\sum_{j \in N}\varepsilon_j^2}{|N|} > \frac{1}{|T|}\sum_{i \in T}(\theta_i^*)^2\right) \leq \exp\left(-\frac{|N|\max\left(1, \frac{\sum_{i \in T}(\theta_i^*)^2}{|T|} - 1\right)}{8}\right).$$

Summarizing these conclusions together, we have with probability at least

$$1 - \frac{2}{n^{4a^2}} - \max\left(\frac{1}{n^{\frac{s}{32}}}, \frac{1}{n^{\frac{(1+a)^2}{16}}}\right) - \exp\left(-\frac{\sum_{i \in T}(\theta_i^*)^2}{72}\right) - \exp\left(\frac{-n^{1-a_0}\sum_{i \in T}(\theta_i^*)^2}{24(2 + \log n)}\right)$$

$$- \exp\left(-\frac{|T|\max\left(1, \frac{\sum_{i \in T}(\theta_i^*)^2}{|T|} - 1\right)}{8}\right) - \exp\left(-\frac{|N|\max\left(1, \frac{\sum_{i \in T}(\theta_i^*)^2}{|T|} - 1\right)}{8}\right),$$

we have $\|\theta(\bar{\tau}) - \theta^*\|_2 < \|\eta(\omega, \lambda) - \theta^*\|_2$ for any $\lambda \geq \sqrt{\log n}$. $\qquad\square$

**Proposition E.3.** *Denote*

$$t_j^* = \min_{t>0} \left\{ t : \int_0^t \frac{\omega(j)}{1+\frac{1}{\beta}} \left( 1 - \exp\left( -\kappa\left(1+\beta\right)\left(\tilde{t} - \frac{1+\beta}{\omega(j)}\right)\right)\right) d\tilde{t} > 0 \right\}.$$

*Then there exists a unique $\{C_1^1, ..., C_1^n\}$ with $C_1^j > 0$ and $\{C_2^1, ..., C_2^n\}$ with $C_2^j > 0$ such that*

- ***Strong Signal Coefficients.*** *For each $j$, when $t > t_j$,*

$$\gamma_j(t) = C_1^j \exp\left( -\frac{\kappa t\left(1 + 2\beta - \sqrt{1+4\beta^2}\right)}{2}\right) + C_2^j \exp\left( -\frac{\kappa t\left(1 + 2\beta + \sqrt{1+4\beta^2}\right)}{2}\right) + \omega(j);$$

(23a)

$$\theta_j(t) = \left( C_1^j + \frac{1}{\kappa\beta}\left( -\frac{\kappa t\left(1 + 2\beta - \sqrt{1+4\beta^2}\right)}{2}\right)\right) \exp\left( -\frac{\kappa t\left(1 + 2\beta - \sqrt{1+4\beta^2}\right)}{2}\right)$$
$$+ \left( C_2^j + \frac{1}{\kappa\beta}\left( -\frac{\kappa t\left(1 + 2\beta + \sqrt{1+4\beta^2}\right)}{2}\right)\right) \exp\left( -\frac{\kappa t\left(1 + 2\beta + \sqrt{1+4\beta^2}\right)}{2}\right) + \omega(j).$$

(23b)

- ***Weak Signal Coefficients.*** *For each $j$, when $t \leq t_j$,*

$$\theta_j(t) = \frac{\omega(j)}{1+\beta} \left( 1 - \exp\left( -\kappa\left(1+\beta\right)\left(t - \frac{1+\beta}{\omega(j)}\right)\right)\right) \text{ and } \gamma_j(t) = 0. \qquad (24)$$

*Proof.* It can be directly checked that that the Eq. 23a, 23b, 24 for any positive $\{C_1^1, ..., C_1^n\}$ and $\{C_2^1, ..., C_2^n\}$. To ensure the continuity of $\theta(t)$ and $\gamma(t)$ (they are continuous since they are differentiable) at $t_j$, we can determine $\{C_1^1, ..., C_1^n\}$ and $\{C_2^1, ..., C_2^n\}$. The uniqueness of $\{C_1^1, ..., C_1^n\}$ and $\{C_2^1, ..., C_2^n\}$ comes from the uniqueness of the solution $(\theta(t), \gamma(t))$, which can be similarly obtained by Picard-Lindelöf Theorem. $\square$

**Theorem E.4.** *Under the same assumptions in Thm. E.2. Suppose $\kappa$ is sufficiently large, then we can obtain the same results in Thm. E.2 for Eq. 5.*

*Proof.* We can know from the solution of Eq. 5 given by Prop. E.3 and the solution of Eq. 3 given by Prop. E.1 that the solution of $(\theta(t), \gamma(t))$ in Eq. 23, 24 is continous with respect to $\kappa$ and converges to the solution of Eq. 3 given by Prop. E.1. Therefore the conclusions in Thm. E.2 hold when $\kappa$ is sufficiently large. $\square$

## F    THEORETICAL ANALYSIS FOR DISCRETE FORM

**Proposition F.1.** *The $\ell_i(\theta(k), \gamma(k)) := \frac{1}{2}(\omega_i - \theta_i(k))^2 + \frac{\beta}{2}(\theta_i(k) - \gamma_i(k))^2$ is non-increasing as long as $\delta < \frac{2}{\kappa \max(1,\beta)}$ in Eq. 6b.*

*Proof.* Denote $H := \nabla^2 \ell_i(\theta(k), \gamma(k)) = \begin{pmatrix} 1+\beta & -\beta \\ -\beta & \beta \end{pmatrix}$. Denote $\begin{pmatrix} \theta_i(k+1) - \theta_i(k) \\ \gamma_i(k+1) - \gamma_i(k) \end{pmatrix} :=$
$\Delta_i(k)$. We have

$$\ell_i(\theta(k+1), \gamma(k+1)) - \ell_i(\theta(k), \gamma(k)) = \langle \nabla \ell_i(\theta(k), \gamma(k)), \Delta_i(k) \rangle + \frac{1}{2} \Delta_i(k)^\top H \Delta_k(k)$$

$$\leq -\frac{1}{\delta} \langle -\delta \nabla \ell(\theta(k), \gamma(k)), \Delta_i(k) \rangle + \frac{\|H\|_2}{2} \|\Delta_i(k)\|_2^2$$

$$\leq -\frac{1}{\delta} \left\langle \begin{pmatrix} \frac{\theta_i(k+1) - \theta_i(k)}{\kappa} \\ \rho_i(k+1) - \rho_i(k) + \frac{\gamma_i(k+1) - \gamma_i(k)}{\kappa} \end{pmatrix}, \Delta_i(k) \right\rangle + \frac{\|H\|_2}{2} \|\Delta_i(k)\|_2^2$$

$$\leq -\frac{1}{\delta} \langle \rho_i(k+1) - \rho_i(k), \gamma_i(k+1) - \gamma_i(k) \rangle + \left( \frac{\|H\|_2}{2} - \frac{1}{\kappa \delta} \right) \|\Delta_i(k)\|_2^2.$$

Since

$$\langle \rho_i(k+1) - \rho_i(k), \gamma_i(k+1) - \gamma_i(k) \rangle = |\gamma_i(k+1)| + |\gamma_i(k)|$$
$$- \rho_i(k) \cdot \gamma_i(k+1) - \rho_i(k+1) \cdot \gamma_i(k) \geq 0,$$

we have $\ell_i(\theta(k+1), \gamma(k+1)) \leq \ell(\theta(k), \gamma(k))$ as long as $\kappa \delta \|H\|_2 \leq 2$. Since $\|H\|_2 \leq \max(1, \beta)$, we have that $\delta < \frac{2}{\kappa \max(1, \beta)}$. $\square$

**Theorem F.2.** *Denote $K := \lfloor \frac{(1 + \frac{1}{\beta}) \bar{\tau}}{\delta} \rfloor$ with $\delta = \frac{1}{\kappa(1+\beta)}$ and $\bar{\tau}$ defined in Thm. D.2. Denote $\theta^*_{\max} := \max_i |\theta^*_i|$. Besides, we inherit the definition $\theta^*_{\max, T}$ in Thm. E.2. For $(\theta(k), \gamma(k))$ in Eq. 6, if $n > 4^{1/(1-a_0)}$ and $\kappa$ is sufficiently large, then with probability at least*

$$1 - \frac{2}{n^{4a^2}} - \max \left( \frac{1}{n^{\frac{s}{32}}}, \frac{1}{n^{\frac{(1+a)^2}{16}}} \right) - \exp \left( -\frac{\sum_{i \in T} (\theta^*_i)^2}{72} \right) - \exp \left( \frac{-n^{1-a_0} \sum_{i \in T} (\theta^*_i)^2}{24(2 + \log n)} \right)$$

$$- \exp \left( -\frac{|T| \max \left( 1, \frac{\sum_{i \in T} (\theta^*_i)^2}{|T|} - 1 \right)}{8} \right) - \exp \left( -\frac{|N| \max \left( 1, \frac{\sum_{i \in T} (\theta^*_i)^2}{|T|} - 1 \right)}{8} \right),$$

*we have*

1. **Strong Signal Recovery.** *For the strong signal coefficients $\theta^{*,s}_S$,*
$$\|\theta_S(K) - \theta^{*,s}_S\|_2 < \|\eta(\omega, \lambda)_S - \theta^{*,s}_S\|_2, \tag{25}$$
   *for any $\lambda > 0$.*

2. **Weak Signal Recovery.** *For the weak signal coefficients and nulls $\theta^*_T$, there exists $\infty > \beta^* > 0$ such that*
$$\|\theta(K)_T - \theta^*_T\|_2 < \|\theta^*_T\|_2 \text{ i.e., } \beta = \infty \text{ Shrinkage to 0;}$$
$$\|\theta(K)_T - \theta^*_T\|_2 < \|\varepsilon_T\|_2 \text{ i.e., } \beta = 0 \text{ No Shrinkage.}$$
   *We can obtain a similar result for $\theta^*_{S^c}$, the weak signal coefficients and null coefficients.*

3. **The Whole Signal Recovery.** *For the whole signal coefficients $\theta^*$, there exists $\infty > \beta^* > 0$ such that*
$$\|\theta(K) - \theta^*\|_2 < \|\eta(\omega, \lambda) - \theta^*\|_2$$
   *for any $\lambda \geq \sqrt{\log n}$. That means Eq. 3 can be more accurate than Minimax and similar approaches in Donoho & Johnstone (1994).*

*Proof.* It is sufficient to prove that for any residue $e > 0$, there exists a $\kappa^o$ such that as long as $\kappa > \kappa^o$, the following condition holds:

$$\gamma_i(k) = 0, \ \forall k \leq K \text{ and } i \in S^c. \tag{26}$$

$$\left| \theta_i(K) - \frac{\omega_i}{1+\beta} \right| < e, \ \forall i. \tag{27}$$

$$|\theta_i(K) - w_i| < e, \ |\theta_i(K) - \gamma_i(K)| < e, \ \forall i \in S. \tag{28}$$

We first prove Eq. 26. With $z_i(k) := \rho_i(k) + \frac{\gamma_i(k)}{\kappa}$ and $z_i(0) = \theta_i(0) = 0$, it follows from Eq. 6 that

$$\frac{z_i(k+1) - z_i(k)}{\delta} = -\frac{\beta}{1+\beta}\left(\frac{\theta_i(k+1) - \theta_i(k)}{\kappa\delta} - \tilde{\varepsilon}_i\right).$$

where $\tilde{\varepsilon}_i := \theta_i^* + \varepsilon_i \ \forall i \in S^c$ and $\theta_i^* = 0$ if $i \in N$. Therefore, we have

$$z_i(k) = -\frac{\beta}{1+\beta}\frac{\theta_i(k)}{\kappa} + \frac{\beta}{1+\beta}k\delta\tilde{\varepsilon}_i.$$

Note that $\ell_i(k) := \ell_i(\theta(k), \gamma(k)) := \frac{1}{2}(\omega_i - \theta_i(k))^2 + \frac{\beta}{2}(\theta_i(k) - \gamma_i(k))^2$ in Eq. F.1 is non-increasing since $\delta < \frac{2}{\kappa\max(1,\beta)}$, therefore, we have

$$|\theta_i(k)| \le |\omega_i| \le \theta_{\max}^* + (1 + a/2)\sqrt{2\log n} \overset{\Delta}{=} B$$

condition on $\max_{1\le j\le n}|\varepsilon_j| \le (1 + \frac{a}{2})\sqrt{2\log n}$. Then we have

$$|z_i(k)| \le \frac{\beta B}{(1+\beta)\kappa} + k\delta\frac{\beta}{1+\beta}\tilde{\varepsilon}_i < \frac{\beta B}{(1+\beta)\kappa} + \bar{\tau}(1+a)\sqrt{2\log n},$$

since $|\tilde{\varepsilon}_i| < \frac{a}{2}\sqrt{2\log n}$ and we have conditioned on $\max_{1\le j\le n}|\varepsilon_j| \le (1 + \frac{a}{2})\sqrt{2\log n}$. Then there exists $\kappa^{(1)} > 0$ such that for any $\kappa > \kappa^{(1)}$, we have

$$|z_i(k)| < \frac{\beta B}{(1+\beta)\kappa} + \bar{\tau}(1+a)\sqrt{2\log n} < 1,$$

for any $k \le K$. This can prove Eq. 26. Next we prove Eq. 27. Note that Eq. 6b is equivalent to:

$$\theta_i(k+1) = \theta_i(k) - \kappa\delta\left((1+\beta)\theta_i(k) - \omega_i\right),$$

which implies

$$\theta_i(k+1) - \frac{\omega_i}{1+\beta} = (1 - \kappa\delta)\left(\theta_i(k) - \frac{\omega_i}{1+\beta}\right).$$

Denote $err_i(k) := \left|\theta_i(k) - \frac{\omega_i}{1+\beta}\right|$, we have $err_i(k) = (1 - \kappa\delta)^k err_i(0)$. Denote $K_e := \min\{k : err_i(k) < e\}$, then we have

$$K_e \le \frac{\log e - \log err_i(0)}{\log(1 - \kappa\delta)} \le \frac{\log e - \log err_i(0)}{-\log 2}$$

$$\le \frac{\log e - \log B}{-\log 2} \ll \lfloor\frac{(1 + \frac{1}{\beta})\bar{\tau}}{\delta}\rfloor = \lfloor\kappa(1 + \frac{1}{\beta})\bar{\tau}\rfloor := K, \tag{29}$$

which will holds for any $\kappa > \kappa^{(2)}$ for some $\kappa^{(2)} > 0$. Finally we prove Eq. 28. Denote $K_{1,i} := \min\{k : |z_i(k)| \ge 1\}$ for $i \in S$. When $k < K_{1,i}$, we have

$$z_i(k) = \delta\beta\sum_{k=0}\theta_i(k).$$

Denote $K_{0,i} := \min\{|\theta_i(k)| > \frac{(1+3a/2)\sqrt{2\log n}}{1+\beta}$, where $|\theta_i(k)| > \frac{(1+3a/2)\sqrt{2\log n}}{1+\beta}$ can be implied by

$$|\theta_i(k) - \frac{\omega_i}{1+\beta}| < \frac{a/2\sqrt{2\log n}}{1+\beta} \overset{\Delta}{=} b$$

for $i \in S$ conditioning on $\max_{1\le j\le n}|\varepsilon_j| \le (1 + \frac{a}{2})\sqrt{2\log n}$. According to Eq. 29, we have

$$K_{0,i}\delta \ll (1 + \frac{1}{\beta})\bar{\tau},$$

when $\kappa > \kappa^{(3)}$ for some $\kappa^{(3)} > 0$. Besides, it follows from the non-increasing property of $\ell_i(\theta(k), \gamma(k))$ that we have

$$\text{sign}(\theta_i(k)) = \text{sign}(\omega_i)$$

for any $k$ once $\theta_i(k) \neq 0$. If this does not hold, then at some $k$, we have

$$\ell_i(\theta(k), \gamma(k)) > \ell_i(\theta(0), \gamma(0)),$$

which contradicts to the non-increasing property. This means $\theta_i(k) = 0$ or does not change the sign once it becomes non-zero. Therefore, the $|z_i(k)|$ is non-decreasing and moreover, if $K_{1,i} > K_{0,i}$, then for any $k < K_{1,i}$

$$|z_i(k)| = \delta\beta \left| \left( \sum_{k=0}^{K_{0,i}} \theta_i(k) + \sum_{k=K_{0,i}+1} \theta_i(k) \right) \right| \geq \delta\beta \sum_{k=K_{0,i}+1} |\theta_i(k)|$$

$$\geq \delta(k - K_{0,i}) \frac{\beta(1 + 3a/2)\sqrt{2 \log n}}{1 + \beta}.$$

Therefore, we have

$$\delta(K_{1,i} - K_{0,i}) \leq \frac{1 + \frac{1}{\beta}}{(1 + 3a/2)\sqrt{2 \log n}} < \bar{\tau}.$$

Since $|z_i(k)|$ is non-decreasing, then once it is greater than 1, we have $z_i(k) = \gamma_i(k)$. Therefore, we have (recall that we define $\Delta_i(k) = \begin{pmatrix} \theta_i(k+1) - \theta_i(k) \\ \gamma_i(k+1) - \gamma_i(k) \end{pmatrix}$ in Prop. F.1)

$$\frac{\Delta_i(k)}{\kappa} = \delta\left( \begin{pmatrix} \omega_i \\ 0 \end{pmatrix} - Hd_i(k) \right),$$

where $d_i(k) := \begin{pmatrix} \theta_i(k) \\ \gamma_i(k) \end{pmatrix}$ and $H$ is defined in Prop. F.1. Denote $\tilde{\delta}_i := \begin{pmatrix} \omega_i \\ 0 \end{pmatrix}$, then multiplying $H$ on both sides, we have

$$Hd_i(k+1) - \tilde{\omega}_i = (I_2 - \kappa\delta H)(Hd_i(k) - \tilde{\omega}_i).$$

Since $\kappa\delta\|H\|_2 < 1$, then we have $I_2 - \kappa\delta H \succeq 0$ and that $\|I_2 - \kappa\delta H\|_2 \leq \frac{\max(1,\beta)}{1+\beta} \overset{\Delta}{=} c < 1$. Denote $\tilde{err}_i(k) := \|Hd_i(k) - \tilde{\omega}_i\|_2$, we have

$$\tilde{err}_i(k) \leq c^k \tilde{err}_i(k).$$

Applying the same technique in proving Eq. 27, we have that $\tilde{err}_i(K) < \frac{e}{2}$ for any $\kappa > \kappa^{(4)}$ for some $\kappa^{(4)} > 0$, which is sufficient to obtain Eq. 28. $\qquad\square$

## G  RECONSTRUCTED NEURAL SIGNALS IN SEC. 6

We visualize the reconstructed signals in Fig. 6. Specifically, if we denote the SWT transformation as $g$, then for the Wavelet Shrinkage that is shown in the left-hand image, we visualize the original signal $y$ (marked by blue), the reconstructed sparse signal $g^{-1}(\theta_\lambda)$ (marked by orange) and the noise $y - g^{-1}(\theta_\lambda)$ (marked by yellow); for our SWDI that is shown in the right-hand image, we visualize the original signal $y$ (marked by blue), the reconstructed strong signal $g^{-1}\theta^s$ (marked by orange) and the weak signal $g^{-1}\theta - g^{-1}\theta^s$ (marked by yellow), and the noise $y - g^{-1}(\theta)$ (marked by purple).

As shown, the sparse signal of the Wavelet Shrinkage shows a large difference from the original signal, especially at the peaks and valleys of oscillations. Therefore, it may miss a lot of information that may be mainly accounted for by the weak signal and the bias due to the threshold parameter. In contrast, the reconstructed strong signal by our SWDI can learn more information due to the ability to remove bias. More importantly, we are pleasantly surprised to find that the reconstructed weak signal shares a similar trend to the original signal. In this regard, it can well capture the pattern encoded in the neural signal. This result suggests that the weak signal, which may refer to the non-burst signal that can encode the conduct effect of the electric field, has a non-ignorable affection on the formation of neural signals. Such an effect, together with the additional information learned by the strong signal of SWDI over the sparse signal of the Wavelet Shrinkage, can well explain the more significant medication response achieved by our method than the Wavelet Shrinkage.

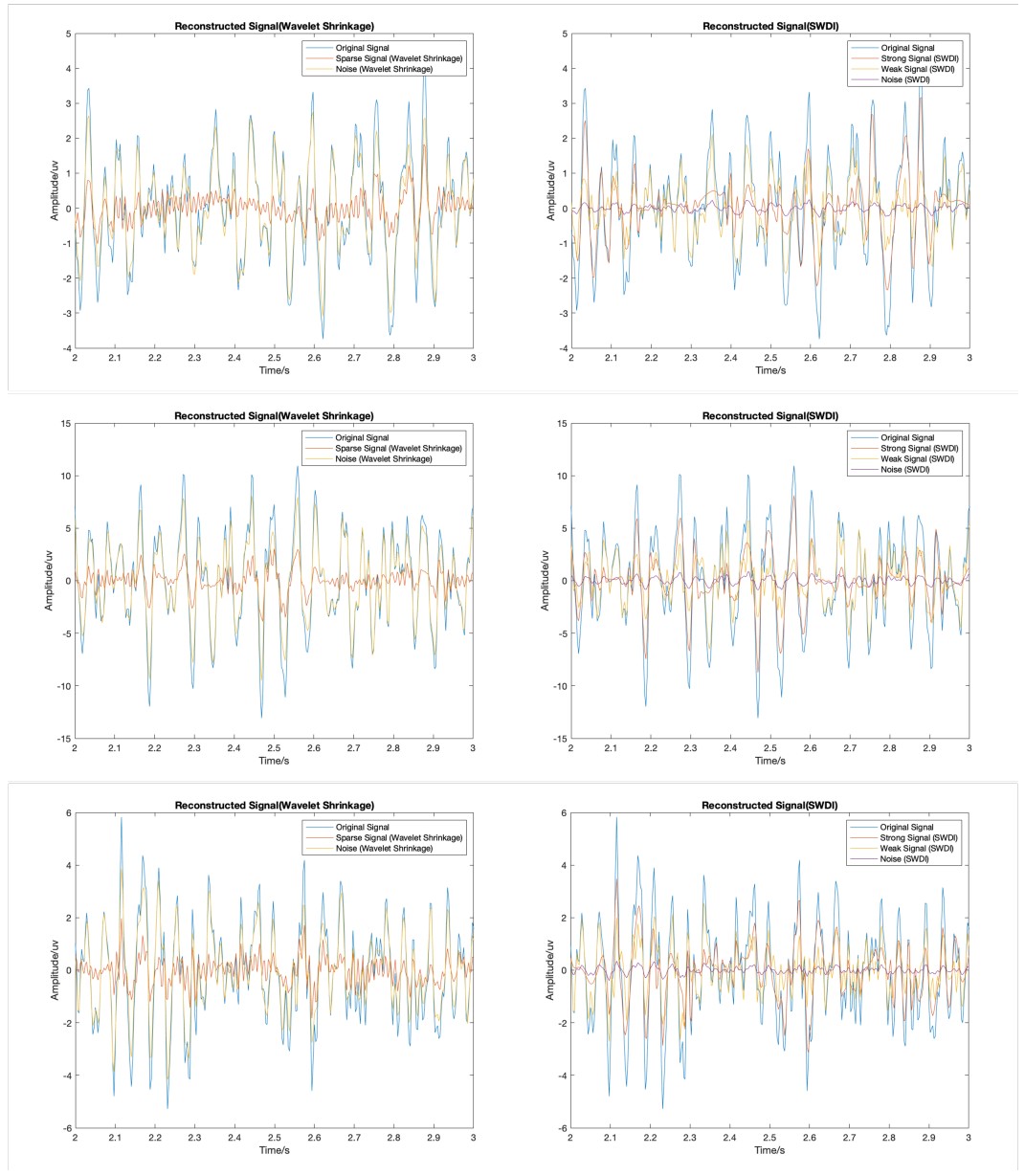

Figure 6: Visualization of reconstructed signals. Left: the original signal $y$ (marked by blue), the reconstructed sparse signal $g^{-1}(\theta_\lambda)$ (marked by orange) by the Wavelet Shrinkage and the noise $y - g^{-1}(\theta_\lambda)$ (marked by yellow); besides the original signal, the right image presents the strong signal $g^{-1}(\theta^s)$ (marked by orange), the weak signal $g^{-1}(\theta) - g^{-1}(\theta^s)$ (marked by yellow) learned by SWDI, and the noise $y - g^{-1}(\theta)$ (marked by purple).

## H  ELECTROENCEPHALOGRAPHY SIGNAL DENOISING

**Data & Problem Description** We extract one subject with 80 trials from Walters-Williams & Li (2011), which comprises a 32-channel Electroencephalography (EEG) signal recorded from a single subject. We added Gaussian white noise to each channel with the signal-to-noise ratio set to 20. We present the mean-squared error (MSE) for the signal reconstruction achieved by our method compared to Wavelet Shrinkage.

**Implementation Details.** Similar to Sec. 6, we perform a $1$-$d$ stationary wavelet transform (SWT) on EEG as the Symlet 8 with level 6. We follow Donoho & Johnstone (1995) to estimate $\sigma$ as

$\tilde{\sigma} = \text{Median}(W_j)/0.6745$. For Wavelet shrinkage, we select $\lambda$ according to the minimax rule in Donoho & Johnstone (1994). For SWDI, we set $\kappa = 20$, $\delta = 1/(\kappa(1+\rho))$ with $\rho = 0.1$ and the stopping time as $\hat{t} = (1 + 1/\rho)/\tilde{\sigma}$.

**Results Analysis.** Fig. 7 shows that our method, especially the dense parameter, can significantly outperform the Wavelet Shrinkage and the sparse parameter, which suggests the importance of capturing weak signals and the capability of our method in learning such weak signals. It's worth mentioning that even when neglecting weak signals, the sparse parameter can still surpass the Wavelet Shrinkage method, thanks to the reduced biases inherent in our methods, as asserted in Thm. D.2 and D.4.

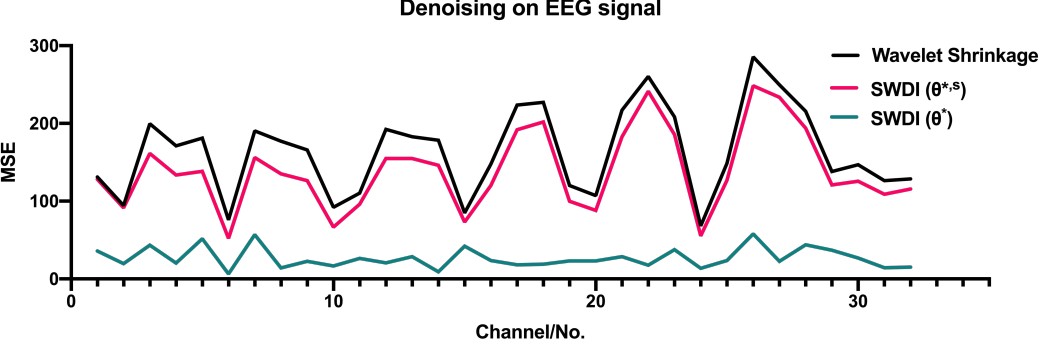

Figure 7: MSE (Microvolts) of the sparse parameter of SWDI $\theta^s(\hat{t})$ (red), dense parameter $\theta(\hat{t})$ (blue) and the Wavelet Shrinkage (black).