# OpenReview forum: "Splitted Wavelet Differential Inclusion"
_ICLR.cc/2024/Conference — Submitted to ICLR 2024_

### Official Review · Reviewer_iHj3 · 2023-10-30

**Soundness:** 2 fair
**Presentation:** 3 good
**Contribution:** 3 good
**Rating:** 5
**Confidence:** 4

**Summary:**

This work proposes to improve classical wavelet shrinkage methods for denoising problems, in order to detect weak signals such as textures which are typically not well recovered in existing methods. By using a framework based on differential inclusion, a method based on l2 splitting and wavelet is proposed, and its theorical properties are analyzed. Application to Parkinson’s disease is also studied.

**Strengths:**

The theoretical results show the advantage of the method compared to classical wavelet shrinkage methods. The results on Parkinson’s disease further provides insights to discover certain activities in signals of scientific interest.

**Weaknesses:**

Certain results need to be further checked, as I find some of them inconsistent. This may be due to some typo but in any case I think the writing should be further improved. The connection between weak signals and textures could also be strengthened.

**Questions:**

-	Check correctness: Theorem 4.3, you said with probability at least something depending on lambda, the eq 2 holds for all lambda. This is quite strange to me. Theorem 4.6, what if the support of T is empty, i.e. |T|=0, does the result still makes sense? What is theta_S^{star,s} in eq. 4?
-	Clarify: what is theta^{star,s} above prop. 3.2? Could you explain show equation 1b in section 4.can give a closed form solution of theta(t)? Is it specific to wavelet transform W? What is the bias you are referring to in Remark 4.2? What is this set {1,4,7,…} in Data synthesis part of Section 5?
-	Typo: statement in prop 3.1, inf over lambda_n instead of lambda? theta_k in eq 6a should be theta(k)?
-	I think it would make more sense to compare with W theta^{star,s} in Fig 1 rather than with the noisy data y on top row. What is the * in the caption of Fig 1? Is it a matrix multiplication or convolution ? How the weak signal lookalike in Fig 5, in relation to textures?

---

### Official Review · Reviewer_Kg8T · 2023-10-30

**Soundness:** 2 fair
**Presentation:** 2 fair
**Contribution:** 2 fair
**Rating:** 3
**Confidence:** 1

**Summary:**

The paper describes a modification to wavelet-based denoising by wavelet coefficient thresholding to account for the presence of weak signals that are usually removed by these methods. The proposed method uses differential inclusion to provide for a gradual accounting of lower-magnitude wavelet coefficients that transitions them from noise (e.g., removing them) to the weak signal as the iteration epoch advances.

**Strengths:**

The paper uses real-world data for testing by studying the correlation between the obtained signal components and diagnostic/medical indicators for the data tested. Analytical results for the estimation performance of the proposed method are provided.

**Weaknesses:**

Given the conditions of the signal components and the relevance of the "weak signal", it does not seem that wavelet shrinkage/denoising is appropriate for this problem. Wavelet denoising is optimal for piecewise smooth signals, and there are no examples in the paper shown to assess if this model is indeed a good match to the "weak signal" that the authors are looking to preserve via wavelet coefficient thresholding.

Related to this concern is that all methods used in the comparison are based on the wavelet decomposition.

There is no discussion of how to distinguish the contribution of weak signals versus noise.

The description of differential inclusion lacks detail. An optimization problem to be solved, or the modification to the solver, is not stated. The role of the function $$\rho$$ introduced is not clear. The theorems state how large a coefficient needs to be to be estimated accurately by the proposed method, but there is no discussion as to whether this guarantee is informative for cases of interest.

Several instances of notation (e.g., $$\theta^{*,s}$$, $$a$$ in Theorem 4.3) are not defined in advance.

**Questions:**

Given that the theoretical results state that coefficient estimates will change from nonzero to zero for weak components, how would a practitioner decide that the iterative algorithm should be stopped? In other words, when will we know that every weak signal component has been accounted for?

Can the performance review include a comparison to approaches from the literature for this problem that do not rely on wavelet transforms? Are there comparable approaches to segment the signal into "strong" and "weak" components?

---

### Official Review · Reviewer_G1Zr · 2023-11-06

**Soundness:** 2 fair
**Presentation:** 2 fair
**Contribution:** 3 good
**Rating:** 5
**Confidence:** 3

**Summary:**

The authors introduce an alternative to wavelet shrinkage whose objective is to recover meaningful yet weak contributions of a signal, given noisy observations

**Strengths:**

The result is interesting in very specific settings, such as e.g. the detection of Parkinson’s Disease and there is some novelty regarding the math. The fact that you go below the noise on T (i.e. the "Weak Signal recovery bounds in Theorem 4.6.") is interesting but it should be better explained and quantified (see my comments below).

**Weaknesses:**

My main concern is with (1) the difficulty to make the distinction between low energy signal and noise. I.e. what do you label as noise and what do you label as “weak signal” ? The motivation seems a little weak to me. For any approach, however fine, there will always be a small amount of noise or a small meaningful signal that you won’t be able to estimate and (2) the fact that the comparison to the soft thresholding approach could be minimal (i.e. despite the strict inequality which is interesting I have to say, If I'm not wrong, there is no intuition on how much of an improvement we get)
Generally speaking, the paper lacks clarity and has to be rewritten. The figures are too small and there are way too many details in the probabilities that appear in the main results (see my detailed comments below). I'm open to discussion but there is some work to be done.

**Questions:**

A couple of general comments:

- From the very beginning of the paper, you talk about differential inclusion but never properly introduce the concept. This makes the whole paper unclear, plus isn’t a differential inclusion a system of the form dx/dt \in S for some S ? where the inclusion is defined on the derivative, and not on the function ?
- Try to simplify your mathematical statements as much as possible. You want them to convey a message as clearly as possible. For the moment, you provide too many details
-We don’t really know by how much you can improve the simple soft thresholding estimator.
-If your error bounds hold for every t>\overline{tau} you should clearly say it

Intro, page 1 and 2

-End of the page: “it is desired to identify the strong signal” —>  “it is desirable”


Page 3
- From what I understand, W is your inverse wavelet transform (I think it would be more clear to sate it like this, even though W might be equal to W^{-1} since the transform is orthogonal)
- To me it does not really make sense to consider zero coefficients if you add noise. How can you make the distinction between coefficients vanishing because of the noise and because they are naturally meaningless ?
- I would add a sentence before Proposition 3.2. E.g. “considering small coefficients does not affect the minimax threshold”
- Also, there is a problem with your statement of Proposition 3.2., doesn’t the minimax error depend on  the level of noise? if there is no noise, how can the minimax error be zero for small coefficients?
- What is theta^{*, s} ? from what I understand this is the part of theta that is left after retaining only the coefficients from S?
- Below the statement of Proposition 3.1. you say that the Donoho and Johnstone estimate is biased because of the non zero lambda. What about vanishingly small lambda’s ?
-The sentence “disentangle the strong signal apart” is not clear. Do you mean recovering the strong signal from the measurements ? or extract this signal from the measurements?

Page 4
- Formulation (1) does not look like the formulation in [5] to me
- I’m not sure I understand proposition 4.1. It seems you never recover theta^* ? I.e the best solution you get is $omega_j$ which is the noisy part?
- The paragraph below Proposition 4.1. is unclear. I might be missing something but the gradient is not the same thing as the bias. In your explanation, I feel there is some confusion between the gradient and the bias. What is the point of ending with theta_j = omega_j if omega_j is noisy ?
- “that different from” —> “that unlike”
- In the statement of Theorem 4.3. Is it for every t>\overline{tau} or does the inequality only hold at one specific time ?

Page 5
- You keep mentioning that when the modulus of rho(t) is one, the (distributional) derivative is zero yet you never explain this in detail. From 3b, it is not clear to me why a modulus of 1 implies a vanishing derivative
- “MAP” stands for maximum a posteriori not maximum a posteriori probability. Btw you should remove this line, this is a well known fact and given how short you already are regarding space, I would avoid losing space unnecessarily
- In the statement of Theorem 4.6., again you lose space unnecessarily by completely expanding the details of your probabilities. Hide this inside asymptotic notations and keep the details for the appendices.
- In the statement of Theorem 4.6. I find the notation 1-a_0 a little dangerous. If a0 can be arbitrarily close to 1 I think you should just remove it as it is upper bounded by the fourth term anyways, it is not really meaningful

Theorem 4.6.
-The use of parentheses is not clear in the 7th term
- Again, does the error bound hold for one \overline{tau} or for all t>tau ?
- The second item is not clear. Do the two bounds hold simultaneously ? Then why not use || \overline{theta}(tau) - theta_T^* || < min(…) ?
- I also have a problem with the general bound on theta(\overline{tau}). You show that your estimator does better than soft thresholding. I give you that. But how well? it is not even clear if it is a minor or a major improvement. Is there a multiplicative constant ? What does this constant depend upon ?
- Is theta^{*,s} the same as theta^*_S ? This is not clear
- In (4), why can you say that |theta^s - theta^{*,S}| = |theta_S - theta^{*,s}_S|, what is theta^{*,S} ? is this the same as theta^{*,s} ?
- Also, after the statement of Theorem 4.6. you claim that you better estimate the components on T. Again, this is not clear to me.Why does the fact that you improve over the rho = 0 or rho = infty imply that you do better regardless of the value of rho ? I.e how can you guarantee that there is no value of rho/lambda for which the soft thresholding approach will recover a better bound than yours?

---

### Author Response · Authors · 2023-11-22
**Response to reviewers**

We thank the reviewers for their efforts in reviewing our paper and constructive feedback. We are sorry for the unclear descriptions that may cause confusion. In the following, we address the main concern regarding the definition and the motivation of weak signals.

The concept of weak signals has been common in statistics, which refers to features that are weakly correlated to the response variable. The contrasting concept of it is strong signals, which have magnitudes above the rate of $O(log(p))$ according to $\ell_1$ regularization [3]. We borrow this definition as the traditional wavelet smoothing method with the soft threshold is equivalent to $\ell_1$ regularization. On the other hand, weak signals are different from random noise since the latter has zero expectation and no correlation with the response variable. Therefore, weak signals have larger coefficients, which align with the definition in [2]. In the scenario of Wavelet Shrinkage, the Universal method has been criticized [4] for learning only strong signals with large coefficients. The goal of this work is to fill in this gap by capturing weak signals, which have shown promising results in our experiments. Particularly interesting, such weak signals correspond to non-burst components in Parkinson's analysis, which aligns with recent findings.




[1] Barber, Rina Foygel, and Emmanuel J. Candès. "A knockoff filter for high-dimensional selective inference." (2019): 2504-2537.

[2] Zhao, Bo, et al. "Msplit lbi: Realizing feature selection and dense estimation simultaneously in few-shot and zero-shot learning." International conference on machine learning. PMLR, 2018.

[3] Zhao, Peng, and Bin Yu. "On model selection consistency of Lasso." The Journal of Machine Learning Research 7 (2006): 2541-2563.

[4] Atto, Abdourrahmane M., Dominique Pastor, and Grégoire Mercier. "Wavelet shrinkage: unification of basic thresholding functions and thresholds." Signal, image and video processing 5 (2011): 11-28.

---

### Meta-Review · Area_Chair_crYp · 2023-12-06

**Metareview:**

The paper proposes an approach to estimate both strong- and weak-components of a (wavelet-sparse) signal contaminated with Gaussian noise. The reviewers raised several issues surrounding the problem formulation, most notably that there is a lack of clarity on how the authors distinguish the contribution of "weak signals" from noise. There was also the issue that the only methods of comparison were other wavelet-based thresholding methods. Overall, the recommendation is a (unanimous) reject.

**Justification For Why Not Higher Score:**

Average sentiment (among the reviewers, as well as my own reading of the manuscript) was negative.

**Justification For Why Not Lower Score:**

N/A

---

### Decision · Program_Chairs · 2024-01-16

Reject